# *PointRePar*: SPATIOTEMPORAL POINT RELATION PARSING FOR ROBUST CATEGORY-UNIFIED 3D TRACKING

**Juntao Liu[1*], Zikun Zhou[1*], Zhoutao Tian[1✉], Guangming Lu[1], Jun Yu[1], Wenjie Pei[1,2✉]**
[1]Harbin Institude of Technology, Shenzhen
[2]Pengcheng Laboratory
wenjiecoder@outlook.com,tianzhuotao@hit.edu.cn

## ABSTRACT

3D single object tracking (SOT) remains a highly challenging task due to the inherent crux in learning representations from point clouds to effectively capture both spatial shape features and temporal motion features. Most existing methods employ a category-specified optimization paradigm, training the tracking model individually for each object category to enhance tracking performance, albeit at the expense of generalizability across different categories. In this work, we propose a robust category-unified 3D SOT model, referred to as SpatioTemporal Point Relation Parsing model (*PointRePar*), which is capable of joint training across multiple categories while excelling in unified feature learning for both spatial shapes and temporal motions. Specifically, the proposed *PointRePar* captures and parses the latent point relations across both spatial and temporal domains to learn superior shape and motion characteristics for robust tracking. On the one hand, it models the multi-scale spatial point relations using a Mamba-based U-Net architecture with adaptive point-wise feature refinement. On the other hand, it captures both the point-level and box-level temporal relations to exploit the latent motion features. Extensive experiments across three benchmarks demonstrate that our *PointRePar* not only outperforms the existing category-unified 3D SOT method CUTrack significantly, but also compares favorably against the state-of-the-art category-specified methods.

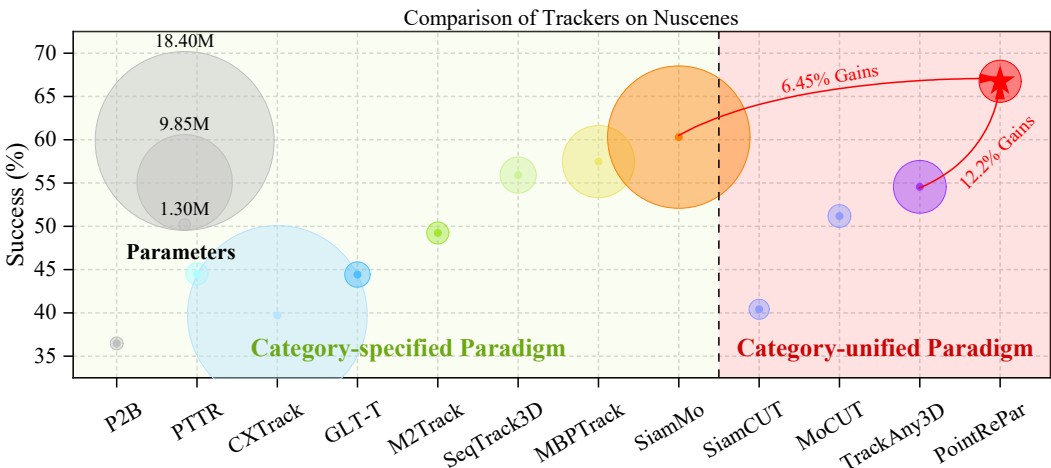

Figure 1: Comparisons on Success of the NuScenes(Caesar et al., 2020) dataset between our method and state-of-the-art 3D trackers. Our approach outperforms the existing category-unified trackers(Nie et al., 2024; Wang et al., 2025) considerably and compares favorably against the category-specified trackers.

---

[*]Equal contribution to this work. ✉ Corresponding authors.

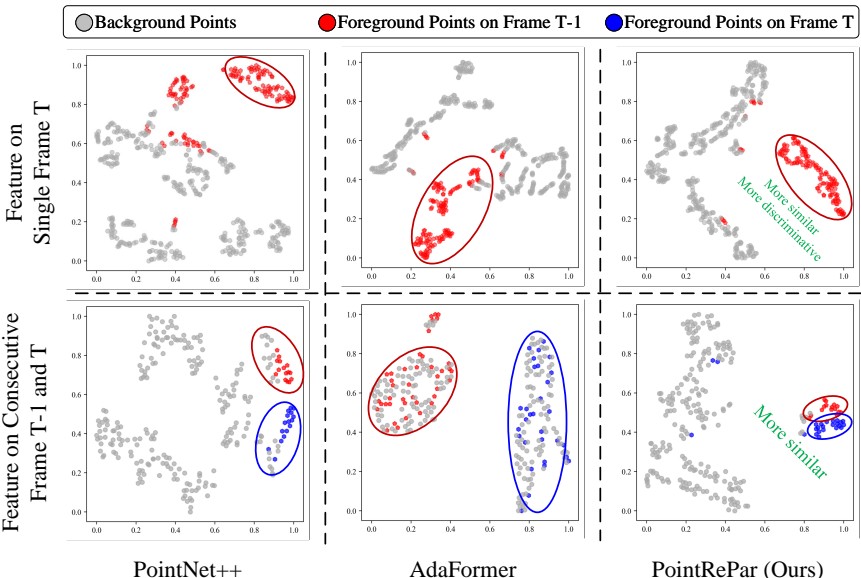

Figure 2: t-SNE maps of encoded features using PointNet++, AdaFormer and our *PointRePar*. Our model is able to not only learn more discriminative spatial features between foreground and background for a single frame (Right Top), but also preserve better temporal feature consistency between two consecutive frames (Right Bottom).

# 1 INTRODUCTION

3D single object tracking (SOT) on point cloud data aims to locate a target object by predicting its position and pose in a streaming way. As a fundamental task in 3D vision, it has a wide range of applications, including autonomous driving and surveillance systems. Despite the remarkable progress of SOT in the 2D visual domain, 3D SOT remains highly challenging due to the inherent crux in effectively learning representations for both spatial shape and temporal motion, arising from the sparsity and incompleteness of point clouds.

Most existing methods(Giancola et al., 2019; Qi et al., 2020; Zheng et al., 2021; Shan et al., 2021; Nie et al., 2023; Xu et al., 2023a;b; Lin et al., 2024; Yang et al., 2024; Zhou et al., 2022; Shan et al., 2021; Liu et al., 2024; 2025; 2023) adopt a category-specified training paradigm, where models are trained separately for each category to ease optimization and improve tracking performance. This paradigm allows the model's entire capacity to focus on learning patterns specific to a single category, significantly reducing the complexity of model fitting and accelerating optimization convergence. However, this learning paradigm suffers from two potential limitations. First, it impedes the tracking model from leveraging and learning the generalizable patterns across categories, thereby limiting its inter-category robustness and generalizability in diverse scenarios. Second, such a paradigm is significantly less efficient than the category-unified training paradigm performing joint training across multiple categories.

Recent studies have initiated the exploration of the category-unified training paradigm for the 3D SOT task. A prominent example is CUTrack(Nie et al., 2024), which introduces deformable grouping vector attention mechanism to dynamically adapt to objects of diverse sizes and shapes across categories. Despite the effectiveness of its designs, CUTrack remains significantly inferior to state-of-the-art category-specified tracking methods, primarily due to its limited feature learning capabilities for both spatial shapes and temporal motions. We conducted a systematic analysis of the contributing factors underlying the failure of prior category-specified approaches and CUTrack within the category-unified training paradigm, which can be attributed to the following reasons: 1) As shown in Figure 2, both the PointNet++ backbone conventionally used in the category-specified methods and AdaFormer proposed by CUTrack suffer from poor feature space separability between target and background points during single-frame point cloud encoding, leading to confusion with the background. 2) Both backbones exhibit substantial feature discrepancies for the same target across consecutive frames. This significant temporal feature inconsistency impedes robust feature matching. 3) Although CUTrack unifies the motion patterns of diverse categories into a normal

distribution, it basically lacks adequate temporal modeling abilities and a category-agnostic design for temporal relations parsing, which further constrains its performance. In this work, we propose a robust category-unified 3D SOT model, named the SpatioTemporal Point Relation Parsing model (*PointRePar*), which enables joint training across multiple categories while excelling in unified feature learning for both spatial shapes and temporal motions of diverse categories.

As shown in Figure 2, our method exhibits superior capability for learning both discriminative spatial shape features in a single frame and temporally consistent motion features between two consecutive frames. Our contributions can be summarized as follows:

- We propose *PointRePar*, an effective yet efficient category-unified 3D SOT framework capable of joint training across multiple categories, delivering excellent tracking performance owing to its sophisticated parsing of spatiotemporal point relations.
- We devise a novel Dynamic Feature Aggregation mechanism for point-wise adaptive feature refinement, upon which the specially adapted 'U-shaped Spatial Relation Parsing Mamba' is further designed to enable our model to capture intricate multi-scale spatial point relations.
- We present a simple yet effective long-term temporal relation parsing scheme, which enables our method to capture dual-level category-agnostic features: point-level motion details and box-level trajectory features.
- We introduce the 'Conditional Gaussian Perturbation' scheme to simulate historical prediction errors under different sparse scenes, thereby enhancing our model's robustness.
- Extensive experiments on KITTI, NuScenes and WOD demonstrate that our model not only outperforms existing category-unified 3D SOT Trackers significantly, but also compares favorably against the state-of-the-art category-specified 3D SOT methods.

## 2 RELATED WORK

### 2.1 3D SINGLE OBJECT TRACKING

Early 3D single object tracking (SOT) approaches(Giancola et al., 2019; Qi et al., 2020; Zheng et al., 2021; Shan et al., 2021; Nie et al., 2023; Xu et al., 2023a;b) inherited the Siamese matching paradigm from 2D SOT. SC3D(Giancola et al., 2019) utilizes Kalman filtering to heuristically sample target proposals and computes similarity with the given template target, selecting the highest-scoring proposal as the predicted result. P2B(Qi et al., 2020) integrates SiamRPN with a point-based Region Proposal Network(Qi et al., 2019) (RPN) for predicting the target bounding box in 3D point clouds, improving tracking efficiency and performance. Most subsequent studies follow this Siamese matching paradigm. BAT(Zheng et al., 2021) introduces box-aware features to enhance the correspondence between the template points and the target points within the search region. PTT(Shan et al., 2021) incorporates a Point-Track-Transformer module into the P2B framework to refine the point features. STNet(Hui et al., 2022) designs a coarse-to-fine correlation network for robust correlation learning. GLT-T(Nie et al., 2023) leverages a global-local Transformer voting scheme to generate high-quality 3D proposals. MTMTrack(Li et al., 2023) first leveraged an encoder-decoder structure which exploits continuous historical boxes as motion prior for coarse box autogressive generation, but its computational overhead and model complexity limit practical deployment.

Recently CXTrack(Xu et al., 2023a) employs a target-centric transformer to model contextual information. MBPTrack(Xu et al., 2023b) designs a memory network and a box-prior localization network to improve CXTrack. M$^2$Track(Zheng et al., 2022) introduces a motion-centric paradigm, modeling the target motion across two consecutive frames for position estimation. Seqtrack3D(Lin et al., 2024) proposes a Seq2Seq framework that combines Siamese and motion-centric paradigms. BEVTrack(Yang et al., 2025),P2P(Nie et al., 2025),SiamMo(Yang et al., 2024) further models the target motion in a Siamese network, achieving state-of-the-art performance. All above methods employ a category-specified training paradigm. Against this background, CUTrack(Nie et al., 2024) introduces a category-unified training paradigm for the first time and proposes dynamic grouping vector-attention as the solution. TrackAny3D(Wang et al., 2025) utilizes a mixture of geometry experts to learn distinct geometric characteristics across diverse obeject categories. However, CUTrack and TrackAny3D suffer from suboptimal computational efficiency, and both performance lag behind category-specified methods. Unlike them, we propose a category-unified 3D tracker that is equipped with mamba, efficient dynamic feature aggregation mechanism and a long-term temporal relation parsing scheme for modeling spatiotemporal point relations robustly. As far as we know, *PointRePar* is the first category-unified 3D tracker that performs favorably against state-of-the-art category-specified trackers.

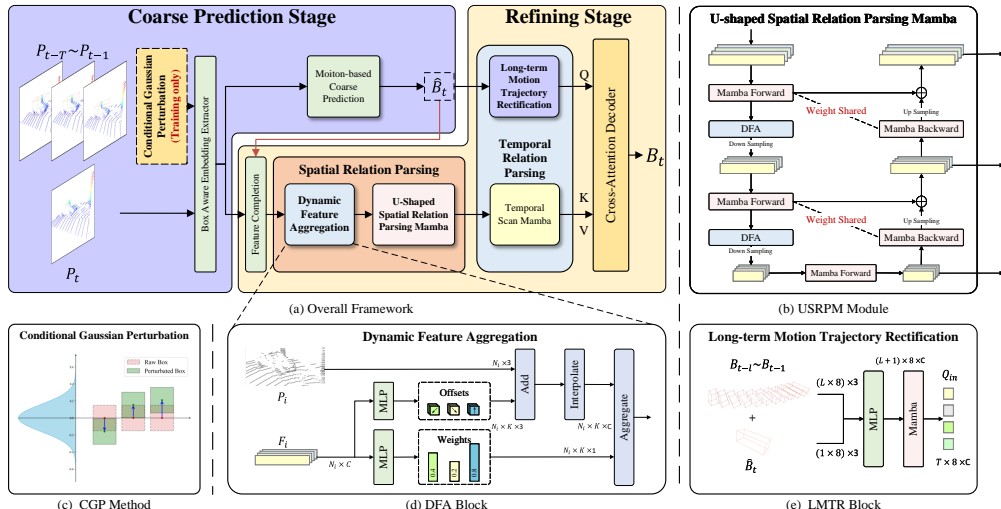

Figure 3: An overview of our proposed *PointRePar* architecture. *PointRePar* first utilizes a Motion-Based lightweight tracker to predict a coarse box in the first stage. In the second stage, *PointRePar* employs a spatial relation parsing backbone to capture the local and global features of point clouds. After that, the Temporal Scan Mamba and Long-term Motion Trajectory Rectification are introduced to perform temporal relation parsing on top of the point features and box sequences. Finally, the enhanced features and box sequence are fed into a cross-attention decoder to predict high-quality boxes.Conditional Gaussian Perturbation is only used in training phase.

## 3 METHOD

### 3.1 OVERVIEW

**Problem Definition.** Given a 3D bounding box (BBox) indicating the initial state of the target object, the task of 3D single object tracking aims to locate the target object by predicting its 3D BBox $\mathcal{B}_s = (x_t, y_t, z_t, \psi_t)$ in a streaming way. Herein, $(x_t, y_t, z_t)$ denote the 3D coordinates of the object at time $t$, and $\psi_t$ refers to its yaw angle. Note that the size of the BBox is assumed to stay unchanged during tracking while the pitch angle and roll angle of the object are always 0, aligning with established conventions in previous studies(Qi et al., 2020; Zheng et al., 2022; Lin et al., 2024).

**Overall framework.** Our *PointRePar* is crafted to capture and parse intricate point-level relations within spatial and temporal domains, aiming to characterize the shape and motion of the target object accurately for category-unified 3D object tracking. As illustrated in Figure 3, *PointRePar* adopts a coarse-to-fine tracking framework, in line with methods(Zheng et al., 2022; Lin et al., 2024). Initially, it utilizes a lightweight tracker consisting of SegPointNet and miniPointNet(Qi et al., 2017a) to efficiently provide a coarse prediction of the object BBox. In the subsequent refining stage, the coarse prediction serves as the query to perform fine-grained decoding through cross-attention, applied to the encoded features enriched with parsed spatiotemporal point relations.

The proposed *PointRePar* leverages Mamba(Gu & Dao, 2023) as the fundamental component for spatial feature encoding, benefiting from its robust long-term sequence modeling capabilities. Specifically, *PointRePar* first leverages a point-based backbone equipped with multi-scale Mamba and the Dynamic Feature Aggregation mechanism to conduct more discriminative and temporal consistent features extraction in category-unified training paradigm. The backbone captures unified multi-scale spatial dependencies through a specially designed 'U-shaped Spatial Relation Parsing Mamba', which hierarchically parses and fuses spatial relations across scales using a Mamba-based U-Net architecture with DFA modules.

To capture category-agnostic long-term temporal relations, *PointRePar* not only models the point-level spatiotemporal relations with 'Temporal Scan Mamba', but it also performs object motion trajectory rectification over the coarse prediction by learning the box-level sequential relations. Furthermore, we introduce the 'Conditional Gaussian Perturbation' scheme, which adds Gaussian noise

to the target object's historical motion trajectory to simulate prediction errors, thereby improving the robustness of the model. In the following subsections, we will elaborate on these designs.

## 3.2 MULTI-SCALE SPATIAL RELATION PARSING

Comprehensive spatial point relation parsing in category-unified training paradigm requires effective modeling of both global and multi-scale features for diverse categories. Therefore, we propose a U-shaped Spatial Relation Parsing Mamba (USRPM), inspired by MSVMamba(Shi et al., 2024), which balances global spatial modeling and computational efficiency. However, integrating Mamba into point set networks alone fails to sufficiently capture local geometric details or adapt to diverse target shapes. To address this, we introduce Dynamic Feature Aggregation (DFA), a module that adaptively aggregates local geometric features across variable receptive fields, enabling robust spatial relation parsing for diverse categories. Details are as follows.

**U-Shaped Spatial Relation Parsing Mamba.** As illustrated in Figure 3(b), the proposed U-Shaped Spatial Relation Parsing Mamba consists of multiple Mamba blocks with Dynamic Feature Aggregation modules and hierarchical downsampling layers. At each level, the point cloud features are processed and progressively downsampled to construct a multi-scale representation. We innovatively design weight-shared Mamba blocks that perform bidirectional scanning during downsampling and upsampling operations. This design not only enables computationally efficient bidirectional modeling as MSVMamba does, but also ensures cross-scale semantic consistency across divergent scanning orientations. The final multiscale feature set $\mathcal{F} = \{F^1, F^2, F^3\}$ aggregates the hierarchical context for the tracking task. Please see more details in the Appendix B.

**Dynamic Feature Aggregation.** Figure 3(d) illustrates the pipeline of Dynamic Feature Aggregation (DFA). Given an input point cloud $\{p_i\}_{i=1}^N \in \mathbb{R}^{N \times 3}$ and its associated features $\{f_i\}_{i=1}^N \in \mathbb{R}^{N \times C}$, this module first generates an offset scale $\Delta p_i \in \mathbb{R}^{K \times 3}$ (confined to $[0, 1]$) and a weight $w_i \in \mathbb{R}^{K \times 1}$ for each point via two MLP layers. These offsets dynamically adjust the positions of the original points, producing $N \times K$ virtual points with corresponding weights $W^o$:

$$P_i^o = \{p_i + \Delta p_{ik} \cdot R_{\max} \mid k = 1, ..., K\}, \tag{1}$$

$$W_i^o = \{\text{Sigmoid}(\text{MLP}(f_i))\}_{k=1}^K, \tag{2}$$

where $R_{\max}$ denotes the maximum displacement radius, and $K$ is the number of offsets per point. Next, we interpolate features for each virtual point using inverse distance weighting:

$$f_{ik} = \frac{\sum_{j=1}^N w_j(x) f_j}{\sum_{j=1}^N w_j(x)}, \quad \text{with } w_j(x) = \frac{1}{d(x, x_j) + \epsilon}, \tag{3}$$

where $d(x, x_j)$ measures the L2 distance between $x$ and $x_j$. Finally, the original point features are updated by aggregating the features of their $K$ virtual counterparts:

$$p_i' = \text{LayerNorm}\left(f_i + \sum_{k=1}^K w_{ik} f_{ik}\right). \tag{4}$$

The DFA module adaptively adjusts the receptive field for each point, enhancing geometric perception for different targets of diverse shapes and sizes while mitigating information loss during downsampling. More importantly, it can enhances intra-object feature similarity while simultaneously boosting feature discriminability against background points within point cloud representations for diverse categories.

## 3.3 LONG-TERM TEMPORAL RELATION PARSING

The category-agnostic trajectory information is critical for handling sparse or occluded point clouds and mitigating drift in cluttered scenes with inter- or intra-class distractors. While previous multi-frame works(Lin et al., 2024; Liu et al., 2024) focus on the temporal modeling of point features, they often neglect the explicit motion encoded in BBox sequences. Therefore, our method jointly models long-term temporal dynamics for not only point features but also target trajectory while maintaining efficiency through a light-weight mamba-based architecture.

**Point-level motion parsing.** For modeling the temporal dynamics based on multi-scale features, we employ a Mamba block to capture frame-wise temporal relations, named as Temporal Scan Mamba, leveraging its strength in long-sequence modeling, as shown in Figure 3. Increasing the temporal modeling window for point features can enhance tracking performance, but it typically results in excessively high computational costs.

**Box-level motion parsing.** Instead, we propose a lightweight Long-term Motion Trajectory Rectification (LMTR) module that efficiently encodes long-term box sequences. Specifically, given a historical box sequence $\mathcal{B}_L = \{B_{t-L}, ..., B_{t-1}\}$ (where $L > T$), we concatenate it with the coarse prediction $B_t$ to form $\hat{\mathcal{B}}_L = \{B_{t-L}, ..., B_t\}$. Following standard practice(Zheng et al., 2021), we convert boxes to corner representations and use a MLP to project them to tokens $X_L \in \mathbb{R}^{L \times 8 \times C}$, augmented with learnable temporal embeddings $E \in \mathbb{R}^{L \times C}$:

$$\hat{X}_L = X_L + E. \tag{5}$$

To model temporal dependencies across the box sequence, tokenized representations are processed through a Mamba block that effectively captures long-range 1-D sequential patterns. The final output tokens spanning the last $T$ time steps are aggregated to construct the decoder input query $Q_{in}$, where each temporal position $Y_i \in \mathbb{R}^{8 \times C}$ encapsulates frame-specific spatial information. This process is formally defined as:

$$Y = \text{Mamba}(\hat{X}_L), \tag{6}$$

$$Q_{in} = \text{Concat}(Y_{-T}, Y_{-T+1}, ..., Y_{-1}). \tag{7}$$

### 3.4 Conditional Gaussian Perturbation

To enhance the robustness of the model against detector localization errors, conventional approaches(Zheng et al., 2022; Lin et al., 2024) inject uniform noise into training trajectories, disregarding the inherent correlation between error magnitude and scene complexity, thereby limiting the performance in sparse scenes. To address this issue, we first investigate error patterns in real-world detection systems. As shown in Figure 4, we observe that localization uncertainty scales inversely with point cloud density, as sparse scenes provide fewer geometric cues for accurate target localization.

Inspired by the above analysis, we propose Conditional Gaussian Perturbation (CGP), which dynamically modulates perturbation intensity according to scene sparsity. Concretely, given an input box $B_i$ with sparsity ratio $r$ (measured as the reciprocal of point density within the detection region), our method applies axis-wise perturbations through a sparsity-conditioned noise generator:

$$\delta_{x,y,z} = \beta_{x,y,z} \cdot c^{-r} \cdot \mathcal{N}(0,1), \tag{8}$$

where $\mathcal{N}(0,1)$ denotes standard Gaussian noise, $c > 1$ controls the exponential decay rate of noise magnitude with increasing density, and $\beta_{x,y,z}$ determines axis-specific scaling factors.

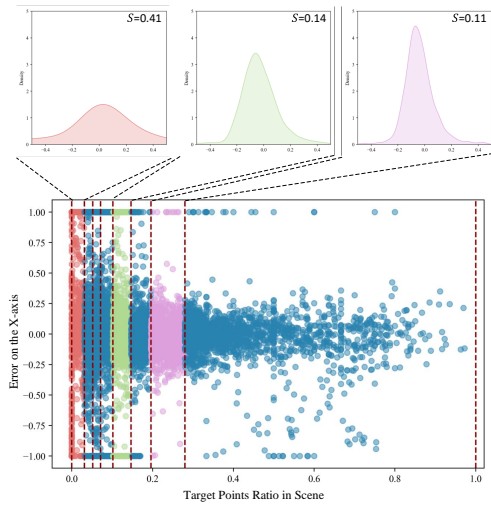

Figure 4: Illustration of the distribution of prediction errors along the X-axis versus the target-to-scene point ratio, defined as the ratio between the number of target points and scene points. The localization uncertainty decreases as point cloud density increases, which implies that sparse scenes offer fewer geometric cues for precise target localization.

By explicitly correlating noise magnitude with scene density, CGP achieves two critical objectives: 1) amplifying perturbations under sparse observations to mimic error accumulation in low-density regions, and 2) preserving physically consistent noise bounds in dense scenarios. This density-aware perturbation strategy, as demonstrated in our experiments, significantly enhances robustness compared to the uniform noise injection.

Table 1: Performance comparisons with state-of-the-art methods in terms of *Success/Precision* on NuScenes and WOD benchmark. "▢" represents models trained in category-unified paradigm and "▢" are trained in category-specified paradigm.

| Method Frame Count | Car [64,159] | Pedestrian [33,227] | Truck [13,587] | Trailer [3,352] | Bus [2,953] | Mean [117,278] |
|---|---|---|---|---|---|---|
| SC3D(Giancola et al., 2019) | 22.31/21.93 | 11.29/12.65 | 30.67/27.73 | 35.28/28.12 | 29.35/24.08 | 20.70/20.20 |
| P2B(Qi et al., 2020) | 38.81/43.18 | 28.39/52.24 | 42.95/41.59 | 48.96/40.05 | 32.95/27.41 | 36.48/45.08 |
| BAT(Zheng et al., 2021) | 40.73/43.29 | 28.83/53.32 | 45.34/42.58 | 52.59/44.89 | 35.44/28.01 | 38.10/45.71 |
| PTTR(Zhou et al., 2022) | 51.89/58.61 | 29.90/45.09 | 45.30/44.74 | 45.87/38.36 | 43.14/37.74 | 44.50/52.07 |
| M²Track(Zheng et al., 2022) | 55.85/65.09 | 32.10/60.72 | 57.36/59.54 | 57.61/58.26 | 51.39/51.44 | 49.32/62.73 |
| MBPT(Xu et al., 2023b) | 62.47/70.41 | 45.32/74.03 | 62.18/63.31 | 65.14/61.33 | 55.41/51.76 | 57.48/69.88 |
| SCVTrack(Zhang et al., 2024) | 58.90/67.70 | 34.50/61.50 | 60.60/61.40 | 59.50/60.10 | 54.30/53.60 | 52.10/64.70 |
| SeqTrack3D(Lin et al., 2024) | 62.55/71.46 | 39.94/68.57 | 60.97/63.04 | 68.37/61.76 | 54.33/53.52 | 55.92/68.94 |
| SiamMo (Yang et al., 2024) | 64.85/72.24 | 46.23/76.26 | 68.22/68.81 | **74.21/70.63** | **65.63/62.07** | 60.31/72.68 |
| PointRePar(Ours) | **71.29/78.20** | **48.36/78.98** | **68.89/71.27** | 71.57/70.56 | 56.40/55.26 | **64.15/76.81** |
| SiamCUT (Nie et al., 2024) | 40.96/44.91 | 31.42/53.80 | 53.91/52.65 | 63.29/58.21 | 41.03/38.01 | 40.41/48.54 |
| MoCUT (Nie et al., 2024) | 57.32/66.01 | 33.47/63.12 | 61.75/64.38 | 60.90/61.84 | 57.39/56.07 | 51.19/64.63 |
| TrackAny3D (Wang et al., 2025) | 59.30/66.46 | 40.37/68.70 | 62.70/62.80 | 66.12/59.20 | 61.01/58.02 | 54.57/66.25 |
| PointRePar(Ours) | **72.95/80.70** | **49.86/79.55** | **75.34/77.82** | **75.87/73.75** | **72.82/72.80** | **66.76/79.64** |

Table 2: Performance comparisons with state-of-the-art methods in terms of *Success/Precision* on KITTI dataset. "▢" represents models trained in category-unified paradigm and "▢" are trained in category-specified paradigm.

| Method Frame Count | Car [6,424] | Pedestrian [6,088] | Van [1,248] | Cyclist [308] | Mean [14,068] | Hardware | Fps |
|---|---|---|---|---|---|---|---|
| SC3D(Giancola et al., 2019) | 41.3/57.9 | 18.2/37.8 | 40.4/47.0 | 41.5/70.4 | 31.2/48.5 | GTX 1080Ti | 2 |
| P2B(Qi et al., 2020) | 56.2/72.8 | 28.7/49.6 | 40.8/48.4 | 32.1/44.7 | 42.4/60.0 | GTX 1080Ti | 20 |
| BAT(Zheng et al., 2021) | 60.5/77.7 | 45.7/72.5 | 52.4/67.0 | 33.7/45.4 | 51.2/72.8 | RTX 2080 | 57 |
| PTTR(Zhou et al., 2022) | 65.2/77.4 | 50.9/81.6 | 52.5/61.8 | 65.1/90.5 | 57.9/78.2 | Tesla V100 | 50 |
| M²Track(Zheng et al., 2022) | 65.5/80.8 | 61.5/88.2 | 53.8/70.7 | 73.2/93.5 | 62.9/83.4 | Tesla V100 | 57 |
| CXTrack(Xu et al., 2023a) | 69.1/81.6 | 67.0/91.5 | 60.0/71.8 | 74.2/94.3 | 67.5/85.3 | RTX 3090 | 29 |
| MBPTrack(Xu et al., 2023b) | 73.4/84.8 | **68.6/93.9** | 61.3/72.7 | 76.7/94.3 | 70.3/87.9 | RTX 3090 | 50 |
| SyncTrack (Ma et al., 2023) | 73.3/85.0 | 54.7/80.5 | 60.3/70.0 | 73.1/93.8 | 64.1/81.9 | TITAN RTX | 45 |
| SCVTrack (Zhang et al., 2024) | 68.7/81.9 | 62.0/89.1 | 58.6/72.8 | 77.4/94.4 | 65.1/84.5 | RTX 3090 | 31 |
| M3SOT (Liu et al., 2024) | 75.9/87.4 | 66.6/92.5 | 59.4/74.7 | 70.3/93.4 | 70.3/88.6 | RTX 3090 | 38 |
| SiamMo (Yang et al., 2024) | 76.3/**88.1** | 68.6/93.9 | 67.9/80.5 | 78.5/94.8 | 72.3/90.1 | RTX 4090 | 108 |
| SiamCUT (Nie et al., 2024) | 58.1/73.9 | 48.2/76.2 | 63.1/74.9 | 36.7/47.4 | 54.0/74.6 | RTX 3070Ti | 36 |
| MoCUT(Nie et al., 2024) | 67.6/80.5 | 63.3/90.0 | 64.5/78.8 | **76.7/94.2** | 65.8/85.0 | RTX 3070Ti | 48 |
| TrackAny3D(Wang et al., 2025) | 73.4/85.2 | 59.6/85.6 | **70.0/82.8** | 74.7/94.0 | 67.1/85.4 | RTX 3090 | 28 |
| PointRePar(Ours) | **76.7/87.5** | **67.4/91.9** | 69.4/82.2 | 74.9/94.1 | **72.0/89.1** | RTX 3090 | 37 |

# 4 EXPERIMENT

## 4.1 EXPERIMENTAL SETTINGS

**Datasets.** To validate the effectiveness of our method, we conduct evaluations on three popular datasets, namely KITTI(Geiger et al., 2012), NuScenes(Caesar et al., 2020) and Waymo Open Dataset (WOD)(Sun et al., 2020). KITTI contains 21 point cloud video sequences with ground-truth labels, spanning 8 object categories. We split the dataset into training, validation, and testing folds following the same setup as previous works(Nie et al., 2024). NuScenes is a more challenging dataset with a large number of sparse scenes. It comprises 700, 150, and 150 scenes for training, validation, and testing, respectively. Following previous works including LiDAR-SOT(Pang et al., 2021) and CUTrack(Nie et al., 2024), we use the WOD dataset solely to evaluate the pretrained models on the KITTI dataset. Specifically, the test set of WOD consists of 1121 tracklets, divided into easy, medium, and hard subsets based on the sparsity of the point clouds.

**Evaluation Metrics.** Following the previous works(Nie et al., 2024; Lin et al., 2024), we use Success and Precision metrics in One Pass Evaluation manner to evaluate our method. Success denotes the Area Under Curve(AUC) for the plot showing the ratio of frames whose intersection over union(IOU) between the ground truth and the predicted BBox is greater than a predefined threshold. Precision is defined as the AUC in terms of distance between the center of two boxes.

**Implementation Details.** Our backbone contains three set abstraction (SA) layers of PointNet++(Qi et al., 2017b) for downsampling, with receptive radius of 0.3, 0.5, and 0.7 meters respectively. We

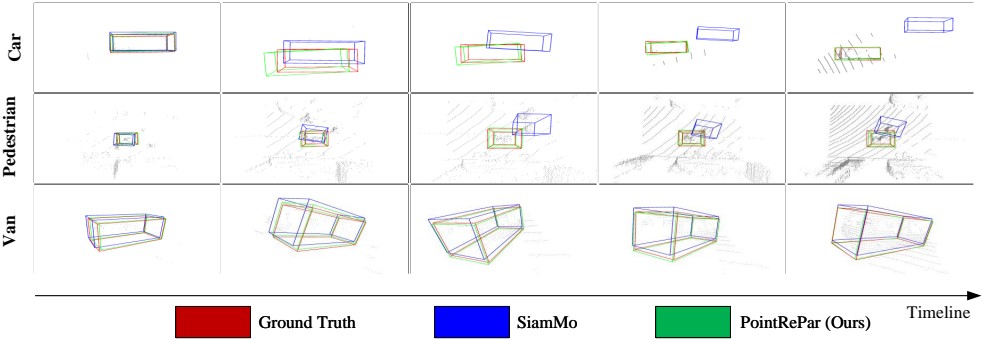

Figure 5: Qualitative comparison between our model and the state-of-the-art category-specified method *SiamMo* on 3 challenging samples with sparse point clouds across categories.

train our model on NVIDIA RTX-3090 GPUs using Adam with a batch size of 64. More training details can be referred to Appendix C.

## 4.2 COMPARISON WITH STATE-OF-THE-ART METHODS

**Results on NuScenes.** Table 1 presents the experimental results on NuScenes dataset containing challenging sparse scenes. It can be observed that our model significantly outperforms MoCUT across all categories, achieving a substantial performance improvement of 15.57/15.01 in terms of Mean Success and Precision, which demonstrates the superiority of our model. Furthermore, compared to previous state-of-the-art category-specified tracking methods, our method also establishes remarkable performance superiority. Moreover, The category-specified PoinRePar still significantly surpasses SiamMo overall, particularly in the large-data categories like "Car", "Pedestrian" and "Truck". Our PointRePar is designed for learning generalizable knowledge, which endows it with distinct performance superiority over SiamMo on large-data categories.

**Results on KITTI.** We compare our proposed *PointRePar* with both category-specified methods and category-unified methods on the KITTI dataset, which are shown in Table 2. Our method surpasses the newest category-unified method MoCUT on most categories, leading to a substantial improvement in 'Mean' performance of both Success and Precision. Furthermore, it is worth noting that our method outperforms most of category-specified methods, only slightly inferior to SiamMo in terms of 'Mean' performance, which is quite encouraging.

**Robustness Comparison on Diverse Sparse Scenes.**

To verify the robustness of *PointRePar* across different sparsity scenes, we compare *PointRePar* with the state-of-the-art method SiamMo in Figure 6, evaluating their performance across different datasets and sparsity conditions. The experiments demonstrate that *PointRePar* achieves overwhelming advantages on the NuScenes dataset and outperforms SiamMo in extremely sparse scenes where the number of target points < 30 on the KITTI dataset, proving the superior robustness of our method compared to category-specific approaches. Appendix E.2 also provides qualitative results in extreme sparse scenes.

**Results on WOD.** To validate the generalization ability of our proposed method across different application scenarios, we follow previous work(Pang et al., 2021; Nie et al., 2024) and conduct an evaluation by applying the models

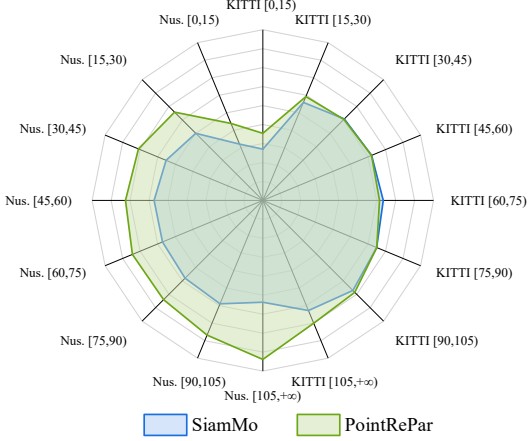

Figure 6: Performance comparison of PointRePar and SiamMo in Success/Precision across varying sparsity levels (represented by different intervals [i, j) of point numbers) on KITTI and NuScenes. 'Nus.' denotes sparse scenes from the NuScenes.

Table 3: Ablation studies of four essential technique designs. We ablate each of these techniques from our *PointRePar* and evaluate the tracking performance on KITTI dataset. DFA denotes Dynamic Feature Aggregation. USRPM denotes U-Shaped Spatial Relation Parsing Mamba. LMTR is Long-term Motion Trajectory Rectification. CGP refers to Conditional Gaussian Perturbation.

| DFA | USRPM | LMTR | CGP | Car [6,424] | Pedestrian [6,088] | Van [1,248] | Cyclist [308] | Mean [14,068] |
|---|---|---|---|---|---|---|---|---|
| | | | | 71.5/85.1 | 60.8/87.2 | 59.8/72.7 | 74.4/94.7 | 65.7/84.6 |
| ✓ | | | | 75.4/86.0 | 61.8/89.1 | 69.3/82.2 | 74.2/94.4 | 68.9/87.2 |
| ✓ | ✓ | | | 76.0/86.3 | 63.0/90.5 | **70.7/83.6** | **77.7/95.1** | 69.9/88.1 |
| | ✓ | ✓ | ✓ | 76.3/86.3 | 60.1/87.3 | 69.0/82.6 | 74.9/94.0 | 68.6/86.6 |
| ✓ | | ✓ | ✓ | 74.8/85.9 | 62.0/88.6 | 68.6/82.3 | 75.5/94.2 | 68.7/87.0 |
| ✓ | ✓ | | ✓ | **76.7**/87.9 | 62.1/89.8 | 69.7/82.9 | 76.5/94.5 | 69.8/88.4 |
| ✓ | ✓ | ✓ | | **76.7**/**88.0** | 63.6/89.6 | 69.1/81.8 | 76.7/94.6 | 70.4/88.3 |
| ✓ | ✓ | ✓ | ✓ | **76.7**/87.5 | **67.4/91.9** | 69.4/82.2 | 74.9/94.1 | **72.0/89.1** |

Table 4: Effect of the number of offset points in the Dynamic Feature Aggregation on four object categories of KITTI dataset.

| Number | Car | Pedestrian | Van | Cyclist | Mean |
|---|---|---|---|---|---|
| **1(Ours)** | **76.7/87.5** | 67.4/**91.9** | **69.4/82.2** | 74.9/94.1 | **72.0/89.1** |
| 2 | 75.5/86.4 | 66.1/90.8 | 67.8/80.8 | 74.4/**94.3** | 70.7/88.0 |
| 3 | 76.3/87.1 | **67.5**/91.6 | 68.9/81.9 | 74.4/94.2 | 71.8/88.7 |

pre-trained on the KITTI dataset to the Waymo Open Dataset (WOD). Our method not only surpasses all other methods in both category-specified and category-unified methods (+1.7%/+5.6% and +4.0%/+9.7% respectively), but also showcases great generalization ability to unseen scenarios. More detailed results can be referred to Appendix D.1.

**Inference Speed.** *PointRePar* achieves an inference speed of 36.6 FPS when running on a single NVIDIA 3090 GPU, which is on par with existing multi-frame methods (e.g. SeqTrack3D). Please refer to the Appendix D.8 for more efficiency details and comparison.

**Qualitative Comparison.** Figure 5 visualizes the tracking results of our *PointRePar* and the state-of-the-art category-specified method *SiamMo* on 3 challenging samples with sparse or incomplete point clouds across different categories. It shows that SiamMo fails to track the target in these challenging scenes and is prone to be misled by intra-class distractors, while our *PointRePar* is able to track the target object consistently and accurately, demonstrating the robustness and reliability of our method in challenging scenarios.

## 4.3 ABLATION STUDIES

**Functionality of essential designs.** In this set of experiments, we conduct ablation studies to investigate the functionality of the essential designs of *PointRePar*. As shown in Table 3, we ablate each of four core designs from our *PointRePar* and evaluate the tracking performance on the KITTI dataset. The results show that all four proposed techniques distinctly enhance the tracking performance, demonstrating their effectiveness. Both the proposed DFA and USRPM benefit most of the object categories as they are designed to capture the spatial point relations and refine shape features.It is noted that without USRPM means that USRPM is replaced with PointNet++. The removal of either one would lead to a significant performance degradation. The proposed CGP is particularly effective for the 'Pedestrian' category, whose motion trajectory is harder to predict than other categories. This is reasonable, as CGP enhances the model's robustness by simulating historical prediction errors. Similarly, LMTR has a positive effect on the 'Car' category which exhibit more variable trajectories.

**Effect of the number of offset points involved in DFA.** To explore the effect of the number of offset points in DFA, we report the results in Table 4. Optimal performance is achieved at **K=1**. Further increasing the number of offset points does not yield additional performance improvements,

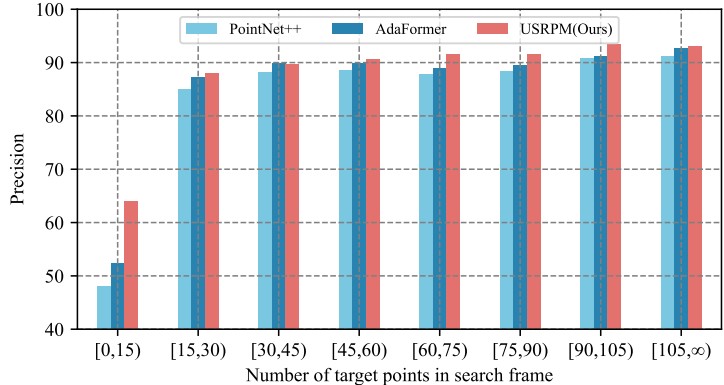

Figure 7: Performance comparison of different backbones on diverse sparse scenes of KITTI.

Table 5: Effect of box sequence length involved in the Long-term Motion Trajectory Rectification (LMTR) on four object categories of KITTI dataset.

| Length | Car | Pedestrian | Van | Cyclist | Mean |
|---|---|---|---|---|---|
| 3 | 73.2/84.4 | 64.7/91.0 | 68.1/80.6 | 73.2/94.1 | 69.1/87.1 |
| 5 | 75.5/86.5 | 66.4/91.7 | 63.1/74.4 | **75.8/94.3** | 70.5/87.9 |
| **7(Ours)** | **76.7/87.5** | **67.4**/91.9 | **69.4**/82.2 | 74.9/94.1 | **72.0/89.1** |
| 10 | 75.2/85.7 | 66.5/**92.2** | 69.0/**82.5** | 75.0/94.1 | 70.9/88.4 |
| 15 | 74.4/85.9 | 64.9/88.8 | 67.0/79.4 | 74.1/94.2 | 68.3/86.8 |

potentially indicating that a single offset point proves sufficient to capture the local geometric context due to the inherent sparsity and singular target nature of 3D SOT point cloud scenarios.

**Effect of USRPM.** To explore the robustness of USRPM in sparse scenes, we report the performance of different backbones on diverse sparse scenes of KITTI in Figure 7. Our USRPM performs favorably against the other backbones in most sparse scenes with minimal computational overhead introduced, particularly in extremely sparse scenes where the number of target points $< 15$, which validates the obvious advantage of USRPM when dealing with sparse scenes. More ablation experiments of USRPM can be referred to Appendix D.4.

**Effect of the box sequence length involved in LMTR.** LMTR is designed to perform object motion trajectory rectification by modeling the box-level sequential relations. We conduct experiments to investigate the effect of the length of box sequence in LMTR on the tracking performance of different categories, set as a hyper-parameter. The results in Tab. 4 show that the overall performance across all categories improves as the box sequence length increases, reaching an optimal value when the length is equal to **7**. Further increasing the sequence length results in performance degradation, potentially indicating that the sequence length has exceeded the trajectory pattern length and is negatively interfering with pattern learning.

## 5  CONCLUSION

In this work, we introduce *PointRePar*, a robust category-unified 3D SOT model. In contrast to typical category-specified 3D SOT methods, our *PointRePar* is capable of joint training across multiple categories to boost the training efficiency substantially while achieving comparable performance to them. *PointRePar* models multi-scale spatial point relations through globally adaptive feature refinement and hierarchical feature parsing, while simultaneously learning long-term motion features by capturing both point-level and box-level temporal relations. Leveraging these essential designs, our method outperforms both category-specified and category-unified 3D SOT methods, demonstrating particular robustness in challenging scenes with sparse point clouds.

## ACKNOWLEDGEMENTS

This work was supported in part by the National Natural Science Foundation of China (Grant No. 62372133, 62125201 and U24B20174), in part by the Guangdong Basic and Applied Basic Research Foundation (Grant No. 2025A1515010705, 2025A1515011546), and in part by the Shenzhen Science and Technology Program (Grant No. JCYJ20240813105901003, KJZD20240903102901003, ZDCY20250901113000001).

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

APPENDIX

## A  DECLARATION OF LLM USAGE

In this study, the Large Language Model (LLM) was employed solely as a supplementary tool for the following limited purposes: Text Polishing: To enhance the fluency and academic formality of expressions without altering original content; Grammar Checking: To identify basic grammatical errors (e.g., tense consistency, subject-verb agreement), excluding logical or terminological modifications. The LLM did not participate in research design, data analysis, or conclusion derivation. Authors retained full control over all academic content.

## B  MORE METHOD DETAILS

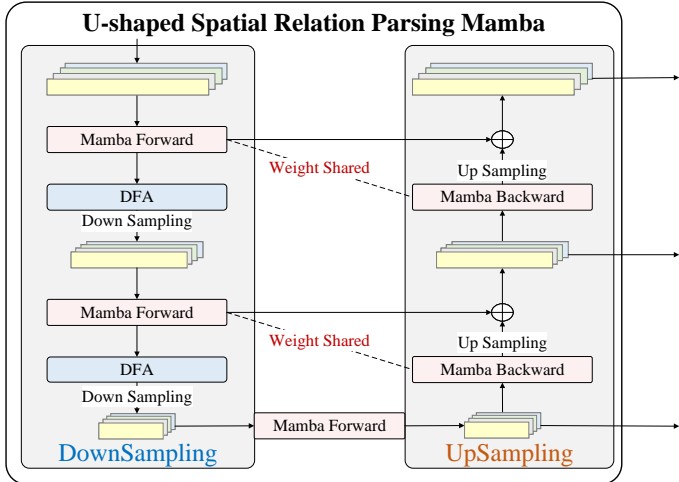

Figure 8: Details of USRPM. It consists of a downsampling path (left branch) and a upsampling path (right branch), outputting multiscale feature which preserves fine-grained spatial details while maintaining semantic consistency.

### B.1  U-SHAPED SPATIAL RELATION PARSING MAMBA

As depicted in Figure 3(b) and Figure 8, USRPM consists of a downsampling path (left branch) and a upsampling path (right branch). Given an input point cloud $\mathcal{P}_i \in \mathbb{R}^{N \times 3}$ and its corresponding features $F_i \in \mathbb{R}^{N \times C}$ at level $i$, the downsampling path (left branch) computes the smaller-scale (low-resolution) features $F_{\text{fwd}}^{i+1}$ and the downsampled points $\mathcal{P}^{i+1}$ as follows:

$$
\begin{aligned}
\hat{F}_{\text{fwd}}^i &= \text{Mamba}^{(i)}(F_{\text{fwd}}^i), \\
F_{\text{fwd}}^{i+1}, \mathcal{P}^{i+1} &= \text{SA}\big(\text{DFA}(\hat{F}_{\text{fwd}}^i), \mathcal{P}^i\big),
\end{aligned}
\tag{9}
$$

where $\text{Mamba}^{(i)}$ denotes the $i$-th Mamba block, $\text{SA}(\cdot)$ is the set abstraction layer from PointNet++ acting as a downsampling operator, and $\text{DFA}(\cdot)$ dynamically aggregates local geometric features. Here, $F_{\text{fwd}}^1 = F_0$ serves as the initial feature input.

In the upsampling path (right branch), high-resolution features are progressively reconstructed by integrating lower-resolution contextual information through skip connections, thereby preserving fine-grained spatial details while maintaining semantic consistency.

For points $\mathcal{P}^i \in \mathbb{R}^{N_i \times 3}$ and features $F_{\text{fwd}}^i \in \mathbb{R}^{N_i \times C}$ at level $i$, we apply the same Mamba block (with shared parameters from the downsampling path) in reverse order. By this way, it achieves knowledge sharing across different scales while maintaining semantic consistency in different order scanning.

$$
\hat{F}_{\text{back}}^{i+1} = \text{Mamba}_{\text{back}}^{(i)}(F_{\text{fwd}}^{i+1}),
\tag{10}
$$

where $\hat{F}_{\text{back}}^{i+1} \in \mathbb{R}^{N_{i+1} \times C}$ denotes the updated feature after reverse scanning. For restoring high-resolution features in the upsampling path, we propagate the updated point features from $N_{i+1}$

points to $N_i$ points with distance-based interpolation method, where $N_i$ and $N_{i+1}$ ($N_{i+1}=N_i/2$) are point set size of input and output of $i$-th downsampling level. The interpolated features $F_{\text{IP}}^i \in \mathbb{R}^{N_i \times C}$ are computed as:

$$F_{\text{IP}}^i = \text{Interpolate}(\hat{F}_{\text{back}}^{i+1}, \mathcal{P}^{i+1}, \mathcal{P}^i), \tag{11}$$

where $\text{Interpolate}(\cdot)$ denotes inverse distance-weighted feature propagation from $\mathcal{P}^{i+1}$ to $\mathcal{P}^i$. These features are then concatenated with the skip-connected forward features $F_{\text{fwd}}^i \in \mathbb{R}^{N_i \times C}$ from the downsampling path and processed through an MLP to refine the fused representation:

$$F^i = \text{MLP}\big(\text{Concat}[F_{\text{IP}}^i; F_{\text{fwd}}^i]\big). \tag{12}$$

At the smallest scale, where no coarser features exist, we set $F^i = F_{\text{fwd}}^i$. The final multiscale feature set $\mathcal{F} = \{F^1, F^2, F^3\}$ aggregates hierarchical context for the tracking task.

## C  MORE IMPLEMENTATION DETAILS

### C.1  INPUT DETAILS

Since SOT only takes care of one target in a scene, the model only needs to consider the subregion in the entire scene where the tracking target may appear for efficiency. Specifically, we scale up the target BBox at (t-1) timestamp by 1 meters to obtain subregions in KITTI and 4 meters in NuScenes. We then sample 1024 points respectively within these subregions to generate the input point clouds. Following Seqtrack3D(Lin et al., 2024), we set time window length to 4 frames, including 3 historical frames and 1 search frame.

### C.2  TRAINING DETAILS

We train our model using the Adam(Kingma, 2014) optimizer with an initial learning rate of 0.0001. The epoch size is set to 100 for KITTI and 40 for NuScenes. In our experimental configuration, the USRPM consists of 3 blocks and each block contains a DFA module and 1 SSM layer with a hidden state dimension of 32. For Temporal Scan Mamba and LMTR, we adopt a 3-layer Mamba block with a hidden state dimension of 16. In addition, the scale factors $\beta_{x,y,z}$ and exponential decay rate $c$ brought by CGP are set to 0.4,0.3,0.2, and 1.5, respectively, considering the training noise distribution of previous methods under normal scenes and appropriately choosing c to enhance noise in sparse scenes.

## D  MORE EXPERIMENTAL RESULTS

### D.1  RESULTS ON WAYMO OPEN DATASET

To validate the generalization ability of our proposed method across different application scenarios, we follow previous work(Pang et al., 2021; Nie et al., 2024) and conduct an evaluation by applying the models pre-trained on the KITTI dataset to the Waymo Open Dataset (WOD)(Sun et al., 2020). The experimental results on WOD are shown in Table 6. We make two notable observations. First, our method consistently surpasses all other methods in both category-specific and category-unified training paradigms in terms of the mean performance of all subsets, which showcases the advantages of our method. Second, both our method and MoCUT outperform all tracking method in the category-specific training paradigm, which reveals the benefits of the category-unified training paradigm in enhancing the model's generalization ability.

Table 6: Performance comparisons with state-of-the-art methods in terms of *Success/Precision* on Waymo Open Dataset (WOD)(Sun et al., 2020). " " represents models trained in category-unified paradigm and " " are trained in category-specified paradigm.

| Method | Vehicle | | | | Pedestrian | | | | Mean |
|---|---|---|---|---|---|---|---|---|---|
| | Easy | Medium | Hard | Mean | Easy | Medium | Hard | Mean | |
| Frame Count | [67,832] | [61,252] | [56,647] | [185,731] | [85,280] | [82,253] | [74,219] | [241,752] | 427,483 |
| P2B(Qi et al., 2020) | 57.1/65.4 | 52.0/60.7 | 47.9/58.5 | 52.6/61.7 | 18.1/30.8 | 17.8/30.8 | 17.7/29.3 | 17.9/30.1 | 33.0/43.8 |
| BAT(Zheng et al., 2021) | 61.0/68.3 | 53.3/60.9 | 48.9/57.8 | 54.7/62.7 | 19.3/32.6 | 17.8/29.8 | 17.2/28.3 | 18.2/30.3 | 33.0/43.8 |
| CXTrack(Xu et al., 2023a) | 63.9/71.1 | 54.2/62.7 | 52.1/63.7 | 57.1/66.1 | 35.4/55.3 | 29.7/47.9 | 26.3/44.4 | 30.7/49.4 | 33.0/43.8 |
| M²Track(Zheng et al., 2022) | 68.1/75.3 | 58.6/66.6 | 55.4/64.9 | 61.1/69.3 | 35.5/54.2 | 30.7/48.4 | 29.3/45.9 | 32.0/49.7 | 44.6/58.2 |
| SiamMo(Yang et al., 2024) | 66.1/73.4 | 58.1/67.9 | 56.6/68.7 | 60.6/70.1 | 44.8/66.2 | 35.6/56.8 | 31.2/51.8 | 37.5/58.6 | 47.5/63.6 |
| SiamCUT(Nie et al., 2024) | 58.3/66.0 | 50.8/60.8 | 49.2/59.1 | 53.0/62.2 | 23.4/36.6 | 20.7/32.0 | 21.4/31.5 | 21.9/33.5 | 35.4/46.0 |
| MoCUT(Nie et al., 2024) | **68.3**/75.0 | **59.4**/66.9 | **57.1**/66.3 | **61.9**/69.7 | 36.5/54.8 | 30.8/48.9 | 29.5/45.4 | 32.4/49.9 | 45.2/58.5 |
| PointRePar (Ours) | 67.3/**78.0** | 58.3/**70.2** | 56.0/**70.3** | 60.9/**73.1** | **45.8/70.8** | **36.5/58.9** | **35.1/58.0** | **40.1/64.4** | **49.2/68.2** |

## D.2 Performance Confidence Intervals

As shown in the Table 7, we add a comparison of IOU and center error 95% confidence intervals. The performance confidence intervals of SiamMo(Yang et al., 2024) and *PointRePar* largerly overlap, which proves that our method is comparable to SOTA category-specific method on KITTI dataset.

Table 7: Confidence intervals of IOU and Center Error on KITTI dataset. IOU refers to the average intersection over union (IOU) between the predicted BBoxes and the ground truth BBoxes. Center Error refers to the average distance between the centers of the two BBoxes.

| Method | IOU↑ | Center Error↓ |
|---|---|---|
| SiamMo | 0.712±0.003 | 0.423±0.028 |
| PointRePar(Ours) | 0.719±0.003 | 0.428±0.029 |

## D.3 The performance gap between the coarse stage and the refine stage

As shown in Table 9 and Table 8, the results reveal that: (1) The distinct performance gap of PoinRepar (Mean Succ./Prec.) between the coarse and refined stages on both datasets indicates the effectiveness of the refined stage. (2) The coarse stage of our PointRePar significantly outperforms the coarse stage of the baseline model, which benefits from the holistic optimization strategy of both the coarse and refined stages through back-propagation.

Table 8: Performance comparison of coarse-stage and refined stage on NuScenes dataset. *Success / Precision* are reported.

| Method | Stage | Car | Ped. | Truck | Trailer | Bus | Mean |
|---|---|---|---|---|---|---|---|
| PointRePar | Coarse | 71.13/79.13 | 48.20/79.09 | 74.13/76.81 | 74.95/73.20 | 71.64/71.61 | 65.22/78.49 |
| | Refine | 72.95/80.70 | 49.86/79.55 | 75.34/77.82 | 75.87/73.75 | 72.82/72.80 | 66.76/79.64 |
| | Gap | 1.82/1.57 | 1.66/0.46 | 1.21/1.01 | 0.92/0.55 | 1.18/1.19 | 1.54/1.15 |

Table 9: Performance comparison of coarse-stage and refined stage on KITTI dataset. *Success / Precision* are reported.

| Method | Stage | Car | Ped. | Van | Cyc. | Mean |
|---|---|---|---|---|---|---|
| Baseline | Coarse | 69.9/83.2 | 59.1/86.5 | 58.8/71.9 | 71.5/93.9 | 64.2/83.9 |
| PointRePar | Coarse | 75.0/86.4 | 64.5/90.9 | 68.1/81.1 | 74.1/93.4 | 69.8/88.0 |
| | Refine | 76.7/87.5 | 67.4/91.9 | 69.4/82.2 | 74.9/94.1 | 72.0/89.1 |
| | Gap | 1.7/1.1 | 2.9/1.0 | 1.3/1.1 | 0.8/0.7 | 2.2/1.1 |

## D.4 Effect of Different Designs in USRPM

To improve temporal feature consistency between consecutive frames, we introduce weight-shared Mamba blocks that perform bidirectional scanning during downsampling and upsampling operations. This design not only enables computationally efficient bidirectional modeling as MSVMamba does, but also ensures cross-scale semantic consistency across divergent scanning orientations. Here, we provide more ablation results to prove the effectiveness of these designs.

**Effect of upsampling path.** Employing upsampling aims to propagate high-level semantic information to low-level features. This assists the model in better leveraging fine-grained features imbued with advanced semantics, thereby benefiting box estimation. As shown in the Table 10, the model without upsampling performs worse in pedestrians, which require richer fine-grained information. Furthermore, the upsampling enables more efficient implementation of bidirectional modeling in Mamba. Employing upsampling would not introduce excessive computational overhead.

**Effect of weight-shared Mamba.** Our USRPM are designed for learning unified multi-scale global feature of diverse categories. Without a weight-shared design, the model fails to achieve knowledge sharing across different scale Mamba block, inducing semantic discrepancies across multi-scale features. As shown in Table 10, our weight-share design demonstrates superior performance.

Table 10: Ablation study of different designs in USRPM. *Success / Precision* are reported.

| | Car | Ped. | Van | Cyc. | Mean |
|---|---|---|---|---|---|
| USRPM | **76.7/87.5** | **67.4/91.9** | **69.4/82.2** | 74.9/94.1 | **72.0/89.1** |
| w/o Upsampling | 76.5/87.4 | 64.2/89.0 | 67.8/80.6 | **75.4/94.2** | 70.4/87.6 |
| w/o Weight-shared | 74.6/85.6 | 63.1/89.7 | 68.3/80.4 | 71.7/93.7 | 69.0/87.1 |

Table 11: Mean average similarity of target features across consecutive frames and between target-background pairs at different scale.

| Backbone | Target to Target | | | Target to Background | | | Performance |
|---|---|---|---|---|---|---|---|
| | $F^1$ | $F^2$ | $F^3$ | $F^1$ | $F^2$ | $F^3$ | |
| AdaFormer | 0.6283 | 0.4589 | 0.2828 | **0.3135** | 0.6426 | 0.8853 | 67.0/86.4 |
| PointRePar-attention | 0.6660 | 0.5750 | 0.6131 | 0.4043 | 0.5033 | 0.4810 | 69.6/87.9 |
| PointRePar-Mamba | **0.6775** | **0.6244** | **0.6342** | 0.3335 | **0.3713** | **0.4108** | **72.0/89.1** |

## D.5 QUANTITATIVE ANALYSIS OF USRPM'S SPATIOTEMPORAL MODELING

Figure 2 demonstrates qualitative results that preliminarily support the hypothesis of *PointRePar*'s effectiveness. To further validate this hypothesis and illustrate how Mamba contributes to the point relation parsing process, we compare our approach with two other attention-based backbones in terms of point feature's spatial discriminability and temporal consistency.

$$AverSim(F_1', F_2') = \frac{1}{n \times m} \sum_{i=1}^{n} \sum_{j=1}^{m} \text{Cosine}(f_i, f_j), \quad where \quad f_i \in F_1', f_j \in F_2',$$

$$meanAverSim_{target-background} = \frac{1}{L} \sum_{i=1}^{L} AverSim(F_O, F_B), \tag{13}$$

$$meanAverSim_{target-target} = \frac{1}{L} \sum_{i=1}^{L} AverSim(F_t^O, F_{t-1}^O),$$

where $F_O$ is the point features of target object and $F_B$ is the point features of background, and $F_t^O$ is the point features of target object at timestamp t.

We employ Formula D.5 to calculate the similarity between the features of target points and background points, as well as between the features of the target in the current frame and the previous frame, at each layer of the backbone network. As shown in Table 11, USRPM demonstrates higher target feature similarity across two consecutive frames, indicating its superior capability in modeling temporal consistency. Additionally, USRPM exhibits lower target-background feature similarity, proving its ability to extract features with stronger spatial discriminability. The quantitative results shown in Table 11 demonstrate that our method achieves superior spatial discriminability and temporal consistency.

## D.6 ADAFORMER VS USRPM

For a fair comparison, we substitute USRPM with AdaFormer in our framework. As shown in Table 12, the experiments results shows that our USRPM is more powerful and efficient. As evidenced in the Table, PointNet++ and AdaFormer proposed by CUTrack(Nie et al., 2024) only achieve 38.2/48.0 and 43.2/52.4, respectively, in sparse scenes where the number of points $< 15$, greatly constraining their overall performance. In contrast, our USRPM performs favorably against the other backbones in extremely sparse scenes with minimal computational overhead introduced. This results also explain why our method outperforms others on NuScenes by a large margin: its

enhanced robustness in sparse scenes is critical, as NuScenes (using 32-beam LiDAR) contains 69% extremely sparse point clouds where the number of points $< 15$, which is significantly more than conventional datasets like KITTI (64-beam LiDAR).

Table 12: Quantitative comparison of different backbones in diverse sparse scenes of KITTI. *Success / Precision* are reported.

| Backbone | [0,15) | [15,+∞) | Overall | FLOPs |
|---|---|---|---|---|
| PointNet++ | 38.2/48.0 | 69.7/89.9 | 65.7/84.6 | 1.80 |
| AdaFormer | 43.2/52.4 | 72.8/91.4 | 67.0/86.4 | 4.52 |
| USRPM(Ours) | **51.9/64.0** | **74.4/92.1** | **72.0/89.1** | 2.42 |

## D.7 ABLATION EXPERIMENTS OF CONDITIONAL GAUSSIAN PERTURBATION

### D.7.1 ABLATION OF THE EXPONENTIAL DECAY RATE IN CONDITIONAL GAUSSIAN PERTURBATION

The exponential decay rate $c$ in CGP controls how rapidly the noise magnitude diminishes relative to the sparsity ratio of the point cloud scene. Excessively high values result in insufficient noise in dense scenes, while overly small values produce excessive noise, which impedes accurate simulation of prediction errors in dense scenes, thereby compromising model's robustness. Therefore, we conduct an ablation experiment to determine the optimal value for the exponential decay rate in CGP. As shown in Table 13, we set the exponential decay rate $c$ to 1.5 in our main experiment.

Table 13: Performance using different exponential decay rate in Conditional Gaussian Perturbation. *Success / Precision* are reported.

| $c$ | Car | Ped. | Van | Cyc. | Mean |
|---|---|---|---|---|---|
| 1.3 | 75.8/86.5 | 65.5/80.7 | 64.3/77.8 | 72.3/93.8 | 70.2/87.7 |
| 1.5 | 76.7/**87.5** | **67.4/91.9** | 69.4/82.2 | **74.9/94.1** | **72.0/89.1** |
| 1.7 | **77.2/87.5** | 62.7/89.5 | **69.6/82.6** | 74.2/93.9 | 70.2/88.1 |
| 2.0 | 74.9/85.5 | 66.1/91.8 | 64.0/76.1 | 73.1/93.9 | 70.1/87.6 |

### D.7.2 EFFECT OF CGP ON SCENES WITH DIFFERENT SPARSITY CHANGING RATE

CGP is designed to add noise perturbations to training samples based on sparsity levels, simulating inference errors to enhance the model's robustness across different sparse scenes. In Table D.7.2, we have conducted an experiments to validate the effectiveness across different sparse change rate intervals [i,j) on KITTI. The results reveal that CGP-trained model still outperforms in scenes with rapid sparsity changes (sparsity rate variation $> 0.4$), proving that CGP can also function effectively in such scenes, which highlights CGP's stability in scenes with rapidly changing sparsity.

Table 14: Performance comparison of model trained with CGP and without CGP in Success/Precision across different sparsity changing rate (represented by different intervals [i, j] of changing rates) on KITTI.

| Method | [0,0,4) | [0.4,1] |
|---|---|---|
| Without CGP | 70.1/88.2 | 68.1/78.7 |
| With CGP | **72.0/89.2** | **69.2/80.0** |

### D.7.3 EFFECT OF CGP ON OTHER METHODS

CGP is a universally applicable training data augmentation method that dynamically adjusts the amplitude of noise perturbation based on point cloud sparsity to generate diverse training samples. Due to its decoupling from specific algorithmic frameworks, CGP can be seamlessly integrated into any existing tracking method. To validate the generalizability of CGP, we conducted experiments by applying it to the siamese-based method BAT(Zheng et al., 2021) and motion-centric method M2Track(Zheng et al., 2022). As demonstrated in Table 15: (1) CGP significantly increases training sample diversity, particularly benefiting underrepresented category (e.g. BAT achieve +33.1/+45.9

improvement on Cyclist.). (2) CGP effectively simulates prediction errors under varying sparsity conditions, thus enhances tracking robustness across all categories. In summary, CGP is not only theoretically compatible with other methods but also practically effective in enhancing their tracking performance.

Table 15: Performance comparisons on the KITTI dataset. *"Improvement"* refers to the performance gain of our CGP training models over the corresponding counterparts.

| Method | Car [6,424] | Pedestrian [6,088] | Van [1,248] | Cyclist [308] | Mean [14,068] |
|---|---|---|---|---|---|
| BAT(Zheng et al., 2021) | 60.5/77.7 | 45.7/72.5 | 52.4/67.0 | 33.7/45.4 | 51.2/72.8 |
| BAT trained with CGP (Ours) | 66.3 / 80.8 | 45.8 / 74.9 | 43.2/65.0 | 66.8 / 91.3 | 55.8/76.6 |
| Improvement | ↑ 5.8 / ↑ 3.1 | ↑ 0.1 / ↑ 2.4 | ↑ 0.8 / ↓ -1.8 | ↑ 33.1 / ↑ 45.9 | ↑ 4.6 / ↑ 3.8 |
| M$^2$Track (Zheng et al., 2022) | 65.5 / 80.8 | 61.5 / 88.2 | 53.8 / 70.7 | 73.2 / 93.5 | 62.9 / 83.4 |
| M$^2$Track trained with CGP (Ours) | 68.3 / 81.2 | 64.4 / 89.7 | 60.3 / 76.3 | 78.1 / 94.8 | 66.1 / 84.7 |
| Improvement | ↑ 2.8 / ↑ 0.4 | ↑ 2.9 / ↑ 1.5 | ↑ 6.5 / ↑ 5.6 | ↑ 4.9 / ↑ 1.3 | ↑ 3.2 / ↑ 1.3 |

### D.8 Efficiency Comparison and Inference Speed

As shown in Table 16, we compared *PointRePar* with other methods in terms of the inference speed, parameter count and computation overhead. Table 17 reports the detailed time consumption. Our *PointRePar* achieves 37 FPS and the average inference time per frame is 27.3 ms. It is noted that, compared to methods (e.g. M$^2$Track(Zheng et al., 2022) and SiamMo(Yang et al., 2024)) utilizing only two consecutive frames, multi-frame methods (e.g. SeqTrack3D(Lin et al., 2024) and *PointRePar*) entails a more time-consuming Pre-process due to the multi-frame input, which explicitly constrains the overall throughput of our framework.

Table 16: Efficiency comparison on the KITTI dataset.

| Method | FPS | Params.(M) | FLOPs(G) |
|---|---|---|---|
| M$^2$Track | 57 | 2.2 | 2.48 |
| CXTrack | 29 | 18.3 | 4.63 |
| MBPTrack | 50 | 7.4 | 2.88 |
| SeqTrack3D | 38 | 3.7 | 5.52 |
| PointRePar(Ours) | 37 | 4.3 | 6.66 |

Table 17: Inference time of each component of our model. TSM refers to Temporal Scan Mamba for point level features. LMTR refers to long-term motion trajectory rectification module for box-level features. CA refers to the cross-attention module.

| Time | Pre-process | First-stage | Feature completion |
|---|---|---|---|
| | 8.8 ms | 2.1 ms | 1.5 ms |
| Time | USRPM | TSM&LMTR&CA | Overall |
| | 8.5 ms | 6.4 ms | 27.3 ms |

### D.9 Mamba vs Attention

We utilize Mamba(Gu & Dao, 2023) as the core component of our *PointRePar* for feature encoding, leveraging its strong long-term sequence modeling capabilities and high modeling efficiency. To investigate the effect of Mamba on our *PointRePar*, we replace all instances of Mamba with the self-attention operation in *PointRePar* and compare the performance of this new variant to the original Mamba-based version. The experimental results presented in Table 18 indicate that the Mamba-based version is computationally more efficient than the self-attention variant, while demonstrating slightly better tracking performance. These observations are consistent with the conclusions drawn from the comparison between them in previous studies(Liang et al., 2024; Han et al., 2024).

## E More Visualizations

### E.1 core modules mechanism visualization

**Visualizations of DFA's receptive field:** Figure 9 visualizes the adaptive dynamic receptive fields of the DFA module for foreground (Car/Pedestrian) and background points. As shown in Figure 9,

Table 18: Comparison on the KITTI dataset between Mamba and self-attention, used as the fundamental component in our *PointRePar*, in terms of tracking performance and efficiency. All results are reported from evaluations on a single GeForce RTX 3090 GPU.

| Components | Performance | Backbone | Enc&Dec | Pre/Post process | Inference speed | FLOPs | #Params. |
|---|---|---|---|---|---|---|---|
| Mamba based | 72.0/89.1 | 8.5ms | 6.4ms | 12.4ms | 36.6FPS | 6.66G | 4.3M |
| Attention based | 69.6/87.9 | 13ms | 8.6ms | 12.4ms | 29.4FPS | 9.31G | 5.6M |

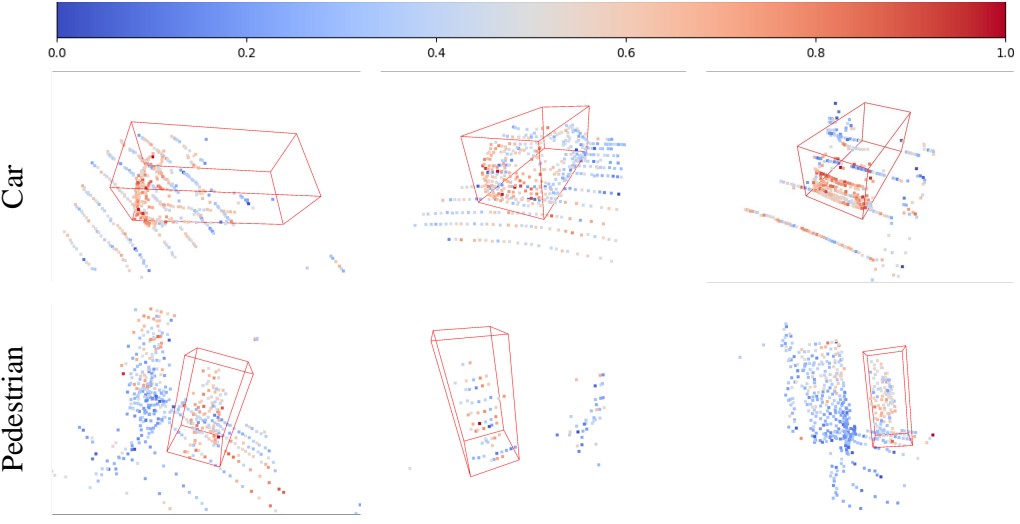

Figure 9: Visualization of the receptive field for a single point (green) in the DFA module. When the green point is a foreground point, its receptive field tends to capture point cloud information (red) from other parts of the target to obtain more shape information, while background points tend to capture similar background points. This phenomenon demonstrates that the DFA can adapt to shapes of different categories and distinguish foreground from background.

Figure 10: Visualization of Mamba's bidirectional scanning in the USRPM module. The stride length (cool-warm colormap) represents the activation value of each point. Mamba assigns larger strides to foreground points, indicating focused attention after bidirectional scanning, which helps mitigate background interference and enhances target localization accuracy.

the foreground points tend to capture holistic target features to acquire structural information while avoiding background confusion, whereas background points focus on contextual information without mixing with foreground points. This behavior demonstrates how DFA enhances spatial feature discriminability.

**Visualization of Mamba bidirectional scanning mechanism:** Mamba dynamically regulates the degree of input information flow into the state space through a parameter called the step size ($\Delta$). A larger step size value indicates that the current token has a more significant impact on the hidden state, which can be interpreted as the token playing a more important role in the sequence. Therefore, in Figure 10 , we visualize the step size values processed by Mamba for each point in point cloud processing. It shows that Mamba assigns larger strides to foreground points (e.g. the front of car), indicating focused attention after bidirectional scanning, which helps mitigate background interference and enhances target localization accuracy.

### E.2 ADDITIONAL QUALITATIVE VISUALIZATIONS

In Figure 11, we provide the tracking results under more scenes featuring extreme sparsity and rapidly changing sparsity. As shown in the Figure 11, regardless of whether it is in extreme sparsity scenarios or rapidly changing sparsity scenes, our method achieves slightly better tracking results compared to SiamMo.

In Figure 12, we provide more visualizations to demonstrate the effectiveness of modeling continuous motion for robust tracking with point clouds. As shown in Figure 12, *PointRePar* achieves a more accurate orientation estimation in Pedestrian category than SiamMo. For car and van object categories, *PointRePar* demonstrates superior localization precision relative to the SiamMo, excelling particularly under sparse point cloud conditions where conventional methods encounter significant estimation challenges.

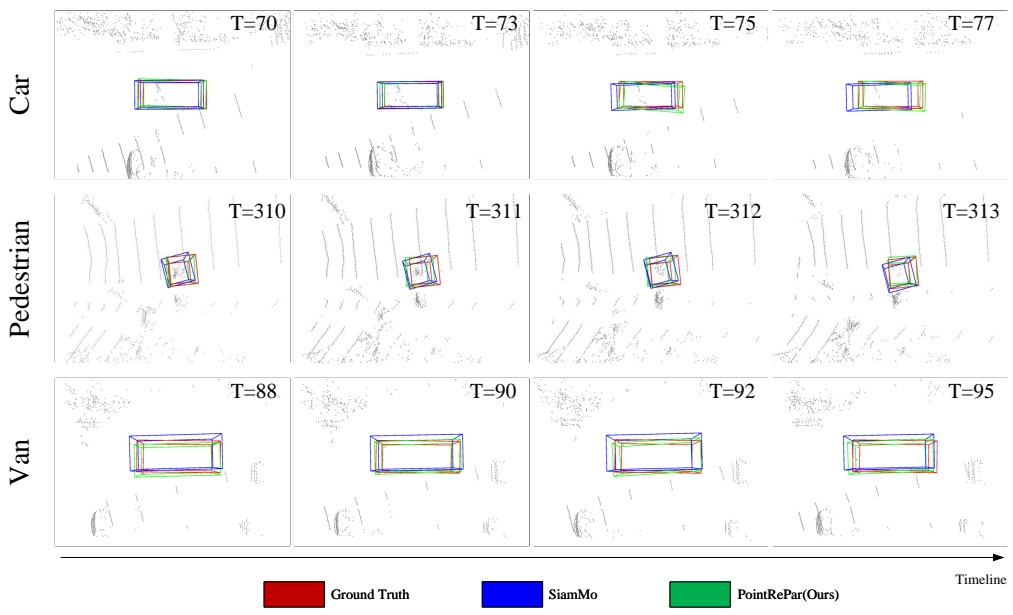

Figure 11: Qualitative comparison between our model and the state-of-the-art category-specific method *SiamMo* in extreme sparse and sparsity fast changing scenes.

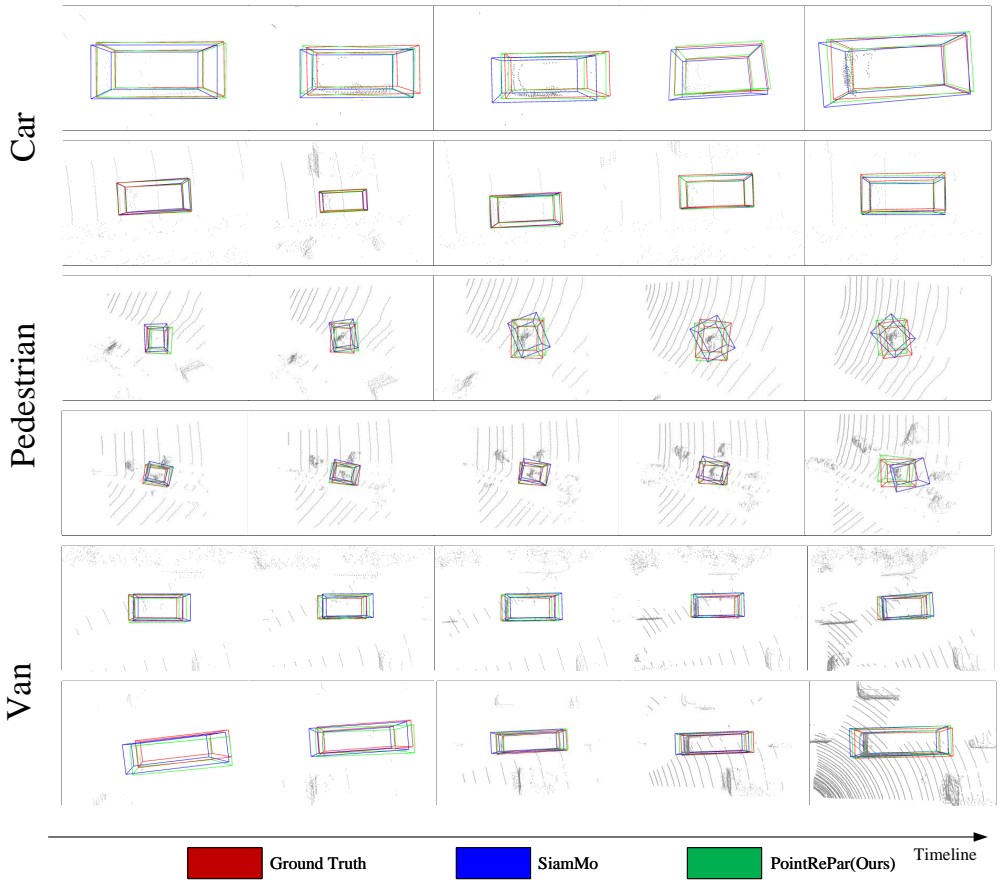

Figure 12: Qualitative comparison between our model and the state-of-the-art category-specific method *SiamMo* on challenging samples across categories.

