# OpenReview forum: "PointRePar : SpatioTemporal Point Relation Parsing for Robust Category-Unified 3D Tracking"
_ICLR.cc/2026/Conference — ICLR 2026 Poster_

### Official Review · Reviewer_pyGc · 2025-10-17

**Soundness:** 3
**Presentation:** 4
**Contribution:** 4
**Rating:** 8
**Confidence:** 5

**Summary:**

This paper proposes the Point Relational Parsing (PointRePar) model, supporting multi-category joint training. Robust tracking is achieved through two core designs: First, the U-shaped Spatial Relational Parsing Module (USRPM) based on Mamba, combined with Dynamic Feature Aggregation (DFA), captures multi-scale spatial point relationships to enhance foreground - background feature separation. Second, it employs dual-level temporal relationship modeling at both point-level (Temporal Scan Mamba) and bounding-box-level (Long-term Motion Trajectory Rectification, LMTR), while introducing Conditional Gaussian Perturbation (CGP) to simulate prediction errors related to scene sparsity. Experiments validated on KITTI, NuScenes, and WOD benchmark datasets demonstrate that PointRePar not only significantly outperforms existing category-unified methods but also rivals current state-of-the-art category-specific approaches. This provides a balanced solution for 3D SOT that optimizes “generalizability, performance, and training efficiency”.

**Strengths:**

1. Existing methods predominantly employ “category-specific training,” requiring separate models for each category, resulting in poor generalization and low training efficiency. Conversely, few unified category methods (such as CUTrack) suffer from insufficient feature discriminative power and weak motion modeling, leading to performance significantly lagging behind category-specific approaches. PointRePar achieves “unified category training + category-specific SOTA performance” for the first time, resolving the core challenge of balancing “cross-category generalization” with “high-performance tracking.”

2. Spatial modeling breaks traditional frameworks: Combining Mamba (high-efficiency long sequence modeling) with a U-shaped structure to design USRPM, paired with DFA for dynamically adjusting point feature receptive fields — — simultaneously resolving the low feature separability issue in PointNet++ and AdaFormer (validated by Figure 2 t-SNE plot) while outperforming Transformer-based methods in efficiency (36.6 FPS inference speed). This achieves a balance between “multi-scale spatial relationship analysis + low computational overhead.”

3. Temporal Modeling Addresses Fine-Grained & Long-Range Requirements: Point-level modeling captures pixel-level motion features between frames, while bounding-box-level LMTR refines trajectories using historical bounding-box sequences. This dual-level design balances “fine-grained motion details” with “long-range trajectory consistency”; CGP dynamically adjusts noise based on scene sparsity, better matching real-world error patterns than traditional uniform noise (Figure 4 validates sparse scene error patterns), significantly enhancing robustness.

4. Strong Module Synergy: The workflow design—“coarse prediction (lightweight tracker) → spatial feature encoding (USRPM) → temporal relationship analysis (TSM+LMTR) → fine decoding (cross-attention)”—forms a complete “spatio-temporal fusion → error robustness → precise localization” chain, with complementary yet non-redundant module functions.

5. Comprehensive Benchmarking: Performance surpasses category-unified methods (MoCUT, TrackAny3D) by over 15%, while matching category-specific SOTA (SiamMo, MBPTrack), challenging the assumption that category-unified approaches inherently underperform category-specific ones.

6. Thorough ablation studies: Validated the effectiveness of four core modules—DFA, USRPM, LMTR, and CGP (Table 3)—analyzed the impact of hyperparameters (DFA offset points, LMTR sequence length) on performance (Tables 4 and 5), and even supplemented efficiency comparisons between Mamba and self-attention (Table 13), yielding highly credible results.

**Weaknesses:**

1. Insufficient analysis of extreme scenarios: While the paper mentions strong performance in sparse scenarios, it lacks dedicated analysis for extreme cases such as “heavy occlusion (target point cloud < 10 points)” or “rapid viewpoint changes (e.g., sharp vehicle turns).” It also fails to compare behavioral differences with SiamMo in such scenarios, making it difficult to fully support the conclusion that “robustness surpasses category-specific methods.”

2. Lack of core module mechanism visualization: Designs like USRPM's Mamba bidirectional scanning and DFA's dynamic receptive field adjustment are validated solely through performance improvements. Without supplementary feature visualizations (e.g., attention weight maps for different-scale features) or error heatmaps, it remains difficult to intuitively demonstrate “how spatio-temporal relationship analysis enhances tracking accuracy.”

**Questions:**

1. Regarding Mamba's advantages over USRPM: Compared to Transformers or RNNs, is there quantitative evidence of Mamba's “long sequence modeling advantage” on unstructured point cloud data? For example, under identical FLOPs, how does Mamba's feature separation (e.g., foreground-background cosine distance) compare to Transformers?

2. Regarding CGP's Generalizability: CGP is designed based on the “scene sparsity - error magnitude” correlation. If scene sparsity changes drastically over time (e.g., when an object is momentarily occluded by a building), can CGP's dynamic noise adjustment still function stably? Can qualitative tracking results for such dynamic scenes be provided?

3. To enhance the literature context and highlight research relevance, it is suggested to add references to related work:
[1] Instance-guided point cloud single object tracking with inception transformer, IEEE TIM 2023;
[2] Revisiting Siamese-Based 3D Single Object Tracking With a Versatile Transformer, IEEE TPAMI 2025.

**Details Of Ethics Concerns:**

No Ethics Concerns

---

> ### Author Response · Authors · 2025-11-26
> **Response to Reviewer pyGc (1/2)**
>
> Thanks for a lot your valuable comments. We have polished our paper according to your suggestions. Here, we number and address questions as follows.
>
> **[Weakness 1] Analysis of extreme scenarios:**
>
> Thanks for your constructive advices. Following the suggestion, we compared PointRePar and SiamMo on diverse sparse scenes of KITTI and Nuscenes datasets, shown in **[Table pyGc-1]** and **[Table pyGc-2].**
>
> **[Table pyGc-1] Performance comparison of PointRePar and SiamMo in terms of Success and Precision across varying sparsity levels (represented by different intervals [i, j) of point numbers) on KITTI.**
>
> | Method | [0,15) | [15,30) | [30,45) | [45,60) | [60,75) | [75,90) | [90,105) | [105,+∞) |
> | --- | --- | --- | --- | --- | --- | --- | --- | --- |
> | SiamMo | 53.5/68.7 | 67.9/84.0 | **70.4**/88.5 | **71.0**/**90.6** | **71.7/91.7** | **72.5/91.9** | 73.6/92.5 | 71.4/90.2 |
> | PointRePar(Ours) | **57.7/71.7** | **69.6/86.2** | 70.2/**88.7** | **71.0**/90.4 | 70.8/91.2 | **72.5**/91.4 | **74.3/92.6** | **75.0/91.7** |
>
> **[Table pyGc-2] Performance comparison of PointRePar and SiamMo in terms of Success and Precision across varying sparsity levels (represented by different intervals [i, j) of point numbers) on the Nuscenes dataset.**
>
> | Method | [0,15) | [15,30) | [30,45) | [45,60) | [60,75) | [75,90) | [90,105) | [105,+∞) |
> | --- | --- | --- | --- | --- | --- | --- | --- | --- |
> | SiamMo | 56.05/70.76 | 65.03/74.87 | 67.61/76.33 | 68.67/75.96 | 68.62/75.48 | 69.01/74.46 | 69.50/75.09 | 66.85/72.57 |
> | PointRePar(Ours) | **61.97/77.01** | **72.92/83.84** | **75.50/84.96** | **76.21/84.66** | **77.22/85.21** | **77.00/83.97** | **78.44/85.60** | **81.91/87.30** |
>
> These results show that:
>
> 1. PointRePar achieves overwhelming advantages on the NuScenes dataset in all kinds of sparse scenes.
> 2. PointRePar significantly outperforms SiamMo in extremely sparse scenes (e.g., <30 points) on the KITTI dataset.
> 3. It is evident that the performance improvement of PointRePar over SiamMo becomes pronounced as the scenes becomes sparser in KITTI.
>
> In conclusion, the robustness of PointRePar significantly surpasses category-specific method SiamMo.
>
> The above discussions are included in L409-L422.  Besides, we also provide qualitative comparison under extreme sparsity conditions in “**Appendix E.2”**, validating the robustness of our method in extreme scenarios.
>
> **[Weakness 2-1] Visualization of the DFA mechanism :**
>
> Thanks for your insightful suggestion, following which we conduct additional visualizations of the DFA module in ‌**Figure 9** in the paper. Specifically, we visualize ‌the dynamic receptive field for the DFA module‌, which empirically demonstrates the robust adaptation of the module objects of varying shapes (e.g., Car, Pedestrian) by dynamically adjusting the local receptive field. The visualizations reveal that the DFA module effectively captures more structural information at each point, thereby enhancing the spatial relationship modeling.
>
> **[Weakness 2-2] Visualization of Mamba bidirectional scanning mechanism:**
>
> Thanks for the suggestion. We would like to clarify that Mamba's SSM framework employs continuous state updates to capture global dependencies without discrete attention weights.
>
> Mamba dynamically regulates the degree of input information flow into the state space through a parameter called the step size (Δ). A larger step size value indicates that the current token has a more significant impact on the hidden state, which can be interpreted as the token playing a more important role in the sequence. In light of this, we visualize the step size values processed by Mamba for each point in point cloud processing in **Figure 10** in the paper to gain more insights of bidirectional scanning mechanism.
>
> **Figure 10** shows that Mamba assigns larger strides to foreground points (e.g. the front of car), indicating focused attention after bidirectional scanning, which helps mitigate background interference and enhances target localization accuracy. We have included the above discussions in **“Appendix E.1”**.

---

> ### Author Response · Authors · 2025-11-26
> **Response to Reviewer pyGc (2/2)**
>
> **[Q1] Quantitative evidence of Mamba's advantage:**
>
> Thanks for your insightful question. We have provided quantitative experiments to compare the sequence modeling performance between Mamba and Transformer in **[Table pyGc-3]**, by measuring the average cosine similarities between foreground and background for evaluation of spatial discriminability, as well as the cosine similarities of target features across adjacent frames for evaluation of temporal consistency.
>
> **[Table pyGc-3] Mean average similarity of target features across adjacent frames at different scale.**
>
> | Method | $F_1$ | $F_2$ | $F_3$ |
> | --- | --- | --- | --- |
> | Adaformer | 0.628 | 0.458 | 0.283 |
> | PointRePar-attention | 0.666 | 0.575 | 0.613 |
> | PointRePar-Mamba | **0.678** | **0.624** | **0.634** |
>
> **[Table pyGc-4] Mean average similarity of features between target-background at different scale.**
>
> | Method | $F_1$ | $F_2$ | $F_3$ |
> | --- | --- | --- | --- |
> | Adaformer | **0.313** | 0.643 | 0.885 |
> | PointRePar-attention | 0.404 | 0.503 | 0.481 |
> | PointRePar-Mamba | 0.334 | **0.371** | **0.411** |
>
> In **[Table pyGc-3]** and **[Table pyGc-4]**, we evaluate the spatial discriminability and temporal consistency of features at different scales from various backbones under identical FLOPs constraints. The experimental results demonstrate that:
>
> - ‌**Spatial Discriminability**‌: The Mamba-based backbone achieves superior feature separation (e.g., lower cosine similarity between foreground/background points) compared to attention-based backbones (e.g., AdaFormer and our attention-based baseline).
> - ‌**Temporal Consistency**‌: Mamba exhibits stronger robustness in maintaining feature coherence across consecutive frames, as evidenced by higher feature similarity of the same target across consecutive frames.
> - The attention-based PointRePar also incorporates the DFA mechanism, shows higher spatial discriminability compared to Adaformer, further validating the effectiveness of DFA.
>
> In conclusion, the quantitative results demonstrate that PointRePar can model features with stronger spatial discriminability and temporal consistency.
>
> We have included above discussion in **Appendix D.5**‌.
>
> **[Q2] Generalizability of CGP:**
>
> Thanks for your insightful comment.
>
> CGP is designed to add noise perturbations to training samples based on sparsity levels, simulating inference errors to enhance the model’s robustness across different sparse scenes.
>
> To address your concern, we have conducted an experiment to validate the effectiveness across different sparse change rate intervals [i,j) on KITTI in **[Table pyGc-5]**.
>
> **[Table pyGc-5] Performance comparison of Model trained with CGP and without CGP in Success/Precision across different sparsity changing rate (represented by different intervals [i, j) of changing rates) on KITTI.**
>
> | Method | [0, 0.4) | [0.4, 1] |
> | --- | --- | --- |
> | Without CGP | 70.1/88.2 | 68.1/78.7 |
> | With CGP | **72.0/89.2** | **69.2/80.0** |
>
> The results reveal that CGP-trained model still outperforms model without CGP in scenes with rapid sparsity changes (sparsity rate variation >0.4), proving that CGP can also function effectively in such scenes, which highlights CGP’s stability in scenes with rapidly changing sparsity.
>
> We have included the above discussion in **Appendix D.7.2** and provided qualitative results in **Figure 11**.
>
> **[Q3] Regarding the literature context and research relevance:**
>
> We appreciate your valuable suggestions. Relevant citations have been properly added to the manuscript.
>
> Please let us know if you would like any further details. Once again, we sincerely appreciate your valuable feedback and look forward to further discussion with you.

---

> > ### Comment · Reviewer_pyGc · 2025-11-27
> >
> > Thanks for your thoughtful response. My concerns have largely been addressed, and I will maintain my positive rating.

---

### Official Review · Reviewer_ncKM · 2025-10-29

**Soundness:** 3
**Presentation:** 4
**Contribution:** 3
**Rating:** 6
**Confidence:** 4

**Summary:**

This paper presents PointRePar, a category-unified 3D single object tracking (SOT) method that leverages spatiotemporal point relation parsing to achieve robust tracking across multiple object categories. The authors propose a Mamba-based U-Net architecture for multi-scale spatial relation modeling, a dynamic feature aggregation (DFA) mechanism, a long-term temporal relation parsing module, and a conditional Gaussian perturbation (CGP) scheme to enhance robustness. Extensive experiments on KITTI, NuScenes, and Waymo Open Dataset demonstrate that PointRePar outperforms existing category-unified methods and is competitive with state-of-the-art category-specific trackers.

**Strengths:**

1. The paper introduces a unified tracking framework that effectively combines spatial and temporal relation parsing using Mamba-based architectures, which is innovative.

2. The paper focuses on the unified category 3D SOT tracking, which has solid contributions. Most 3D trackers are category-dependent, which is unnecessary and introduces significant redundancy into 3D SOT trackers.

3. The results are comprehensive and good.

**Weaknesses:**

1. The overall method is too complex, which includes a U-Net for feature fusion, motion-modeling, and dynamic feature aggregation.

2. The experimental results should clearly reveal the effects of each module and the hyperparameters design for the proposed module.

3. The speed comparison and ablation studies should be clearly presented.

**Questions:**

See weakness.

---

> ### Author Response · Authors · 2025-11-26
> **Response to Reviewer ncKM**
>
> We sincerely appreciate your insightful comments on our manuscript. Below, we address each of your concerns in detail:
>
> **[Weakness 1] The complexity of the PointRePar:**
>
> Thanks for your constructive feedback. Our PointRePar is designed as a robust category-unified 3D SOT framework equipped with three essential technical designs, being capable of learning generalizable knowledge across categories with strong robustness in sparse scenes.  Despite its apparent sophistication in design, we would like to emphasize that its efficiency remains highly competitive:
>
> 1. The parameter scale is comparable to most existing methods, with 4.3M parameters (similar to SeqTrack3D’s 3.7M).
> 2. PointRePar demonstrates higher efficiency than the latest category-unified method TrackAny3D, achieving 30% faster inference speed in **Table 2** in the paper.
>
> In our future work, we will explore more streamlined architectures that maintain performance while improving efficiency.
>
> **[Weakness 2] Module analysis and hyper-parameters design:**
>
> Thanks for your constructive comment. We have revised the paper to provide a clearer analysis of each module's contribution and a more detailed explanation of our hyper-parameters choices. Here, we provide clearer ablation results of each module in **[Table ncKM-1]**.
>
> **[Table ncKM-1] Ablation studies of four essential technique designs.**
>
> | DFA | USRPM | LMTR | CGP | Mean |
> | --- | --- | --- | --- | --- |
> |  |  |  |  | 66.1/84.1 |
> | √ |  |  |  | 68.9/87.2 |
> | √ | √ |  |  | 69.9/88.1 |
> | √ | √ | √ |  | 70.4/88.3 |
> | √ | √ | √ | √ | **72.0/89.1** |
>
> The above results clearly demonstrate the contribution of each module.
>
> The hyper-parameters section also has been expanded to explain the rationale behind our selection of key parameters.
>
> These improvements should make the experimental analysis more transparent and reproducible.
>
> **[Weakness 3] Regarding speed comparison:**
>
> Thanks for your valuable suggestion. Following your suggestion, we have reorganized the results presentation to make comparisons clearer. The speed comparison with existing methods has been moved to **Table 2**, where it now appears alongside accuracy metrics for direct comparison.
>
> Please let us know if you would like any further details. Once again, we sincerely appreciate your valuable feedback and look forward to further discussion with you.

---

### Official Review · Reviewer_jhyB · 2025-10-30

**Soundness:** 3
**Presentation:** 3
**Contribution:** 3
**Rating:** 6
**Confidence:** 4

**Summary:**

The paper tackles LiDAR‐based 3D Single Object Tracking (SOT) in a category-unified setting. It proposes PointRePar, a coarse-to-fine tracker that parses spatial and temporal point relations to improve robustness across categories. Experiments on KITTI, nuScenes, and WOD show large gains over prior category-unified trackers and competitive results vs category-specified SOTA.

**Strengths:**

1. Clear motivation for unified tracking and robust spatio-temporal relation parsing; the goal and assumptions are explicitly stated

2. Well-structured architecture: a coherent coarse-to-fine pipeline with motion-based coarse prediction, spatial relation parsing (USRPM+DFA), temporal relation parsing (TSM+LMTR), and a cross-attention decoder.

3. SoTA performance.

**Weaknesses:**

1. Limited novelty, the paper’s novelty is mainly compositional/system-level, not a fundamentally new primitive.

**Questions:**

1. Will CGP works for other methods?

2. Could you clarify how each ablation in Table 3 is instantiated? For example, in the ‘w/o USRPM’ setting, is USRPM simply removed, or replaced with a capacity-matched alternative (e.g., a PointMamba-style block)?”

3. How big is the gap between the coarse-stage predictions and the refined predictions?

---

> ### Author Response · Authors · 2025-11-26
> **Response to Reviewer jhyB (1/2)**
>
> Thanks for your insightful comments. We have polished our paper according to your suggestions. Below are our detailed responses:
>
> **[Weakness 1] Clarification of the novelty:**
>
> Thanks for your constructive feedback. While our primary contribution may not be a ground-breaking primitive innovation, we emphasize that the essential novelty of our PointRePar lies in the effective yet efficient category-unified 3D SOT framework, which excels at learning generalizable knowledge across categories with 3 synergistic technical innovations, leading to substantial performance gains in 3D SOT:
>
> 1. We propose the novel Dynamic Feature Aggregation mechanism (DFA) and structure-adapted Mamba, which enable our PointRePar to capture intricate spatial discriminative and temporal consistent point cloud features.
> 2. Long-term temporal relation parsing scheme (LMTR) is specifically designed category-agnostic trajectory modeling in both point level and box level.
> 3. Conditional Gaussian Perturbation (CGP) is further introduced to enhance the model's robustness in sparse scenes.
>
> Both ablation study and experimental comparison with other methods demonstrate the effectiveness of each functional design and substantial performance superiority over other methods.
>
> **[Q1]** **Will CGP work for other methods?**
>
> Thanks for the insightful suggestion. Following the suggestion, we have conducted experiments to validate the CGP’s effectiveness for two classical methods, i.e., **siamese-based approach BAT** and **motion-centric method M2Track, which** are shown in **[Table jhyB-1]**.
>
> **[Table jhyB-1] Effect of CGP on BAT and M2Track on KITTI dataset.**
>
> | Method | Car | Pedestrian | Van | Cyclist | Mean |
> | --- | --- | --- | --- | --- | --- |
> | BAT | 60.5/77.7 | 45.7/72.5 | 52.4/67.0 | 33.7/45.4 | 51.2/72.8 |
> | BAT trained with CGP | 66.3/80.8 | 45.8/74.9 | 53.2/65.2 | 66.8/91.3 | 55.8/76.6 |
> | Improvement | $\color{green}5.8/3.1$ | $\color{green}0.1/2.4$ | $\color{green}0.8$/$\color{red}-1.8$ | $\color{green}33.1/45.9$ | $\color{green}4.6/3.8$ |
> | M2Track | 65.5/80.8 | 61.5/88.2 | 53.8/70.7 | 73.2/93.5 | 62.9/83.4 |
> | M2Track trained with CGP | 68.3/81.2 | 64.4/89.7 | 60.3/76.3 | 78.1/94.8 | 66.1/84.7 |
> | Improvement | $\color{green}2.8/0.4$ | $\color{green}2.9/1.5$ | $\color{green}6.5/5.6$ | $\color{green}4.9/1.3$ | $\color{green}3.2/1.3$ |
>
> The results show that:
>
> 1. CGP improves the performance of both BAT and M2Track significantly across almost all categories, which demonstrates the generalizable effectiveness of CGP.
> 2.  CGP is particularly effectiveness w.r.t. the low-data categories (e.g. BAT achieve +33.1/+45.9 improvement on Cyclist),  benefiting from its capability of increasing training sample diversity and enhancing the robustness by simulating prediction errors under varying sparsity conditions.
>
> We have included the above discussion into “**Appendix D.7.3”.**
>
> **[Q2] How each ablation in Table 3 is instantiated?**
>
> Thanks for your valuable comments. We sincerely apologize for not clearly specifying these experimental settings in the original manuscript. We have now revised the paper to improve clarity and added more ablation results. In **Table 3**, the ablation settings are implemented as follows:
>
> - ‌**w/o DFA**‌: The DFA module is entirely removed from the backbone network.
> - ‌**w/o USRPM**‌: The USRPM module is replaced with a standard PointNet++ backbone (which includes the DFA module). Additional ablation experiments analyzing the USRPM structure (including replacing a PointMamba-style backbone and attention-based backbone) are provided in the **"Appendix D.4 and D.9"**.
> - ‌**w/o LMTR**‌: The trajectory modeling component is completely disabled, meaning no box sequence modeling is performed.
> - ‌**w/o CGP**‌: The model is trained without our proposed conditional Gaussian perturbation, using only conventional uniform noise distribution as in prior work.

---

> ### Author Response · Authors · 2025-11-26
> **Response to Reviewer jhyB (2/2)**
>
> **[Q3] How big is the gap between the coarse-stage predictions and the refined predictions?**
>
> Thanks for your meticulous comments. Following the suggestion, we provide the performance gap between the coarse stage and the refine stage on Nuscenes and KITTI in **[Table jhyB-2]** and **[Table jhyB-3]** respectively.
>
> **[Table jhyB-2] Performance gap between the coarse and refine stages on Nuscenes dataset.**
>
> | Method | Car | Pedestrian | Truck | Trailer | Bus | Mean |
> | --- | --- | --- | --- | --- | --- | --- |
> | Coarse-Stage (PointRepar) | 71.13/79.13 | 48.20/79.09 | 74.13/76.81 | 74.95/73.20 | 71.64/71.61 | 65.22/78.49 |
> | Refine-Stage (PointRepar) | 72.95/80.70 | 49.86/79.55 | 75.34/77.82 | 75.87/73.75 | 72.82/72.80 | 66.76/79.64 |
> | Gap | 1.82/1.57 | 1.66/0.46 | 1.21/1.01 | 0.92/0.55 | 1.18/1.19 | 1.54/1.15 |
>
> **[Table jhyB-3] Performance gap between the coarse and refined stages on KITTI dataset.**
>
> | Method | Car | Pedestrian | Van | Cyclist | Mean |
> | --- | --- | --- | --- | --- | --- |
> | Coarse-Stage (Baseline) | 69.9/83.2 | 59.1/86.5 | 58.8/71.9 | 71.5/93.9 | 64.2/83.9 |
> | Coarse-Stage (PointRepar) | 75.0/86.4 | 64.5/90.9 | 68.1/81.1 | 74.1/93.4 | 69.8/88.0 |
> | Refine-Stage (PointRepar) | 76.7/87.5 | 67.4/91.9 | 69.4/82.2 | 74.9/94.1 | 72.0/89.1 |
> | Gap | 1.7/1.1 | 2.9/1.0 | 1.3/1.1 | 0.8/0.7 | 2.2/1.1 |
>
> The results reveal that:
>
> 1. The distinct performance gap of PoinRepar (Mean Succ./Prec.) between the coarse and refined stages on both datasets indicates the effectiveness of the refined stage.
> 2. The coarse stage of our PointRePar significantly outperforms the coarse stage of the baseline model, which benefits from the holistic optimization strategy of both the coarse and refined stages through back-propagation.
>
> We have included the above discussion in **“Appendix D.3”.** We appreciate your careful review and hope this clarification addresses your question. Please let us know if you require any further details.

---

### Official Review · Reviewer_h3CE · 2025-11-03

**Soundness:** 3
**Presentation:** 2
**Contribution:** 3
**Rating:** 6
**Confidence:** 2

**Summary:**

PointRePar tackles the task of 3D single object tracking, which involves tracking a given 3D bounding box query throughout a sequence of sparse point clouds, ultimately predicting the center and yaw of the bounding box at each frame. Since this task relies solely on point representations, it requires robust shape modeling and effective utilization of temporal features.
To address these challenges, the paper introduces a Mamba-based spatial and temporal encoder. Moreover, because the inference process is autoregressive, the model must be resilient to errors propagated from previous predictions. To enhance robustness, the authors propose a dynamic Gaussian perturbation strategy during training, where the perturbation scale is adaptively adjusted according to the local point sparsity.
Finally, the paper presents a unified model applicable across multiple object categories, demonstrating strong performance on benchmark datasets.

**Strengths:**

1. Strong performance on the benchmark.

2. An interesting result in the supplementary shows that the model achieves better efficiency and performance when Mamba is used instead of Attention for point cloud processing.

**Weaknesses:**

1. Although category-unified and category-specific paradigms are listed separately in all tables, I believe a fully unified model could still be trained in the same way as the category-specific one. To ensure a fair comparison, PointRePar should be trained separately for each category and compared directly with the state-of-the-art category-specific models. I am not fully convinced that these paradigms are inherently different to the extent that they cannot be fairly compared in the same table.

2. Although Conditional Gaussian Perturbation is only applied during training, Figure 3 may mislead readers into believing that it is also used during inference.

**Questions:**

I am not very familiar with the field of 3D single object tracking, so my evaluation may not be fully conclusive. The paper is overall well-structured and clearly written. However, I am not entirely convinced that the proposed model is truly novel or provides substantial insights to the community.

The use of Mamba for point cloud encoding has already been demonstrated in prior work, such as Zhang et al., “Point Cloud Mamba: Point Cloud Learning via State Space Model,” 2024, as well as in several subsequent studies. While applying Mamba to motion trajectory rectification is an interesting direction, its significance seems limited given that Mamba is already well-known for its strong performance in sequential processing.

Overall, my impression is that the introduction of Mamba forms a major part of this paper’s contribution, but the work does not clearly highlight why or how Mamba is particularly effective for this specific task.

---

> ### Author Response · Authors · 2025-11-26
> **Response to Reviewer h3CE**
>
> Thanks very much for your positive feedback, we have polished our paper according to your suggestions. Please let us know if this resolves your concerns. Below are our detailed responses:
>
> **[Weakness 1] Evaluation of PointRePar in category-specific training paradigm.**
>
> Thanks for your constructive feedback. We agree that a direct comparison with the state-of-the-art category-specific models is valuable. Here, we provide results of PointRePar trained separately for each category in **[Table h3CE-1]**.
>
> **[Table h3CE-1] Performance comparison of SiamMo, category-specified PointRePar on Nuscenes dataset.**
>
> | Method | Car | Pedestrian | Truck | Trailer | Bus | Mean |
> | --- | --- | --- | --- | --- | --- | --- |
> | SiamMo | 64.85/72.24 | 46.23/76.26 | 68.22/68.81 | **74.21/70.63** | **65.63/62.07** | 60.31/72.68 |
> | PointRePar-Specific | **71.29/78.2** | **48.36/78.98** | **68.89/71.17** | 71.57/70.56 | 56.40/55.26 | **64.15/76.81** |
> | PointRePar-Unified | **72.95/80.70** | **49.86/79.55** | **75.34/77.82** | **75.87/73.75** | **72.82/72.80** | **66.76/79.64** |
>
> The results reveal that:
>
> 1. Category-unified PointRePar consistently outperforms both category-specified PoinRePar and SiamMo across all categories, which demonstrates the capability of PointRePar  to learn generalizable knowledge across categories during category-unified training, thereby achieving stronger tracking performance than the category-specific training paradigm.
> 2. The category-specified PoinRePar still significantly surpasses SiamMo overall, particularly in the large-data categories like “Car”, “Pedestrian” and “Truck”. Our PointRePar is designed for learning generalizable knowledge, which endows it with distinct performance superiority over SiamMo on large-data categories.
>
> In the revised version, we have included the above discussion of **[Table h3CE-1]** in **“Table 1”.**
>
> **[Weakness 2] The illustration of Conditional Gaussian Perturbation‌ in Figure 3**:
>
> Thanks for the valuable advice. We have polished Figure 3 and added a corresponding note in the Figure 3 caption to explicitly state that the CGP is only applied during training.
>
> **[Q1] The novelty of Mamba design and the contributions to the community:**
>
> Thanks for your valuable comment.
>
> **[Q1-1] Clarification on Contributions:**
>
> We agree that applying Mamba to point cloud encoding has been explored before and is not our major contribution. Instead, the novelties of our PointRePar lie in three aspects:
>
> 1. We propose a category-unified robust framework for 3D SOT, PointRePar, which achieves SOTA performance.
> 2. DFA module and USRPM are introduced to address the existing key limitations in feature space discriminability and temporal consistency with category-unified 3D SOT.
> 3. Further, LMTR is designed to model category-agnostic trajectory information, while CGP is employed to enhance the model’s robustness in sparse scenes.
>
> Following the constructive comments of the reviewer,  we will reorganize the Introduction Section to highlight the above contributions and eliminate the potential misunderstanding.
>
> **[Q1-2] Customized Adaptations of Mamba for Category-unified 3D SOT**
>
> Existing point cloud encoders used in 3D SOT suffer from insufficient feature space discriminability and temporal consistency. As shown in **[Table h3CE-2]**, direct application of Mamba with multi-directional scanning (like PointMamba-style) fails to resolve these issues. In contrast, we design two core adaptations to Mamba in our PointRePar to tackle these issues, leading to substantial performance gains in **[Table h3CE-2]**:
>
> 1. **Multi-scale Scanning.** To accommodate targets of varying sizes and shapes, we introduce multi-scale scanning to model global information across scales, improving representation consistency.
> 2. **Weight-Shared Bidirectional Scanning.** Existing multi-directional scanning may result in inconsistent feature representations across different scanning orders. Our design employs weight sharing to enhance feature consistency.
>
> **[Table h3CE-2] Ablation study of different designs in USRPM. Success / Precision are reported.**
>
> | Method | Mean |
> | --- | --- |
> | PointNet++ with Mamba | 70.4/87.6 |
> | PointNet++ with BiMamba | 69.0/87.1 |
> | USRPM (Ours) | **72.0/89.1** |
>
> Our designs not only enable robust handling of size-varying objects but also enhance temporal consistency across consecutive frames. Further analysis in **“Appendix D.5”** reveals that our method is capable of modeling features with higher spatial discriminability and greater temporal consistency compared to other structures.
>
> In summary, our work does not merely apply Mamba to 3D SOT but innovatively adapts its architecture to address the task's unique spatiotemporal requirements. We believe this constitutes a meaningful advancement over prior methods.
>
> Once again, we sincerely appreciate your valuable feedback and look forward to further discussion with you.

---

### Author Response · Authors · 2025-12-03
**Summary of Rebuttal (2/2)**

### **More in-depth analysis and detailed clarification**

- **(R-h3CE,  R-jhyB) Clarification of the Novelty and Contributions:**

    We clarify that the essential novelty of our PointRePar lies in the effective yet efficient category-unified 3D SOT framework, consisting of four core technical designs, including 1) the DFA module for point-wise adaptive feature refinement, 2) the specially adapted Mamba called ’U-shaped Spatial Relation Parsing Mamba’ for capturing intricate multi-scale spatial point relations, 3)  the LMTR scheme designed to model category-agnostic trajectory information and 4) CGP enhancing the model’s robustness in sparse scenes. Based on those novel designs, our PointRePar excels at learning generalizable knowledge across categories, leading to substantial performance gains in Category-Unified 3D SOT. Following the constructive comments of the reviewer,  we have reorganized the Introduction Section to highlight the above contributions and eliminate the potential misunderstanding. **(Introduction Section)**

- **In-Depth Analysis and Clearer Presentation:**
    - **(R-ncKM) Method complexity:**

        Efficiency analysis reveals that the parameter scale and efficiency of PointRePar remain highly competitive. **(Appendix D.8)**

    - **(R-pyGc) Lack of core module mechanism visualization:**

        Additional visualizations of core modules are provided to highlight the effectiveness of core modules. **(Appendix E.1)**

    - **(R-h3CE, R-jhyB, R-ncKM) Presentation clarity:**

        We have updated Figure 3 to explicitly state that the CGP is only applied during training and revised the paper with more detailed analyses on the core modules and explanations on the hyper-parameters. **(Table 3)**


With these comprehensive revisions, we have provided detailed responses and extensive new evidence for every point raised by the reviewers and believe most concerns are resolved. We hope this summary facilitates a smooth and efficient review process for the Area Chair. All the revisions or newly added content is marked in blue in the revised manuscript.

---

### Author Response · Authors · 2025-12-03
**Summary of Rebuttal (1/2)**

Dear Area Chair and Reviewers,

We sincerely thank all the reviewers and the area chair for their constructive comments and considerable efforts.  We have carefully addressed all reviewer comments through additional experiments, in-depth analysis and clearer visualizations.  Below, we present a summary of our revisions as follows.

### **Consensus on Strengths**

- **The work is Well-Motivated (R-jhyB, R-pyGc):** PointRePar is well-motivated and addresses the core challenges of the performance gap between category-unified and category-specified 3D SOT.
- **Innovate Framework (R-ncKM, R-pyGc):** PointRePar is recognized to be an effective yet efficient category-unified 3D SOT framework, addressing the existing key limitations in feature space discriminability and temporal consistency within category-unified 3D SOT.
- **Comprehensive Experiments (R-ncKM, R-pyGc):**  Comprehensive results are provided to demonstrate the effectiveness of PointRePar.
- **SOTA Performance (R-h3CE, R-jhyB, R-ncKM, and R-pyGc):** PointRePar achieves state-of-the-art results on challenging benchmarks, not only outperforming existing category-unified methods but also rivaling current state-of-the-art category-specified approaches.

### **Additional Experiments**

- **More Comparative Experiments**:
    - **(R-h3CE) A fair comparison with other category-specified methods:**

        The comparison of category-specified PointRePar against other category-specified methods is provided; **(Table 1)**

    - **(R-jhyB) Require the performance gap between the coarse and the refined predictions:**

        The performance comparison between the coarse and refined stages within PointRePar is provided; **(Appendix D.3)**

    - **(R-pyGc) Insufficient analysis of extreme scenarios:**

        We provide a comparison of PointRePar versus SiamMo across different sparsity levels. **(L409-L422)**


    These analyses further demonstrate the robustness and effectiveness of PointRePar.

- **Generalizability of CGP:**
    - **(R-jhyB) The generalizability of CGP on other methods:**

        We have performed an experiment to investigate the applicability of CGP to other methods. (**Appendix D.7.3**)

    - **(R-pyGc) The generalizability of CGP in scenes with fast changing sparsity:**

        An ablation of CGP in more challenging scenes with different sparsity changing rates is provided. (**Appendix D.7.2)**


    The above experiments demonstrate the generalizability of CGP across different methods in diverse scenes.

- **(R-pyGc) Quantitative Evidence of Mamba's Advantage:**

    Additional quantitative experiments have been conducted to show that the specially adapted Mamba endows PointRePar with distinctly stronger spatial discriminability and temporal consistency in multi-scale features. **(Appendix D.5)**

---

### Author Response · Authors · 2025-12-03

Dear Area Chairs (ACs),

We sincerely appreciate your time and effort in reviewing our paper.

The initial score of our paper is **(6, 6, 6, 8)**. During the discussion period, we made every effort to address the concerns raised by reviewers. So far, we have received positive feedback from Reviewer pyGc that our response largely resolved his/her concerns. Consequently, he/she is willing to maintain the score of 8 in support of the recommendation for our submission. Unfortunately, we have not received any further feedback from other reviewers (i.e. h3CE, jhyB, ncKM) to date.

| Reviewer | Initial Rating | Confidence | Feedback |
| --- | --- | --- | --- |
| h3CE | 6 | 2 | None |
| jhyB | 6 | 4 | None |
| ncKM | 6 | 4 | None |
| pyGc | 8 | 5 | Positive |

**Note:** ‘None’ denotes that we have not received feedback from the reviewer until now.

Best regards,

The authors

---

### Meta-Review · Area_Chair_8gfg · 2025-12-11

**Summary:**

This paper proposes a robust approach for 3D single object tracking (SOT) in a category-unified setting. The method incorporates several carefully designed modules to enhance robustness in sparse scenes, improve feature discriminability, and strengthen temporal consistency. Experimental results demonstrate clear performance gains over both category-specific and category-unified baselines, and all reviewers provided positive scores.

**Reviewer Concerns:**

The primary concern raised pertains to the method’s novelty, given the extensive use of Mamba-based architectures in recent 3D vision research. The authors clarified that the main contributions lie in the newly introduced modules rather than in Mamba itself. Additional minor concerns—including the need for broader comparisons with prior work, more comprehensive ablations, and analysis of the generalizability of the proposed GCP module—were largely addressed in the rebuttal.

**Reviewer Scores:**

Reviewer h3CE gave an initial score of 6 and is inclined to raise it to 8 following satisfactory clarifications. Reviewer jhyB also gave an initial score of 6 and may retain this score due to lingering concerns about novelty despite other issues being addressed. Reviewer ncKM gave an initial score of 6 and is inclined to raise it to 8 after the rebuttal. Reviewer pyGc gave an initial score of 8 and has confirmed maintaining that score.

---

### Decision · Program_Chairs · 2026-01-26

Accept (Poster)